# Value Improved Actor Critic Algorithms

**Yaniv Oren**
Department of Intelligent Systems
Delft University of Technology
2628 CD Delft, The Netherlands
`y.oren@tudelft.nl`

**Moritz A. Zanger**
Department of Intelligent Systems
Delft University of Technology
2628 CD Delft, The Netherlands
`m.a.zanger@tudelft.nl`

**Pascal R. van der Vaart**
Department of Intelligent Systems
Delft University of Technology
2628 CD Delft, The Netherlands
`p.r.vandervaart-1@tudelft.nl`

**Mustafa Mert Çelikok**
Dept. of Mathematics & Computer Science
University of Southern Denmark
Odense, Denmark
`celikok@imada.sdu.dk`

**Wendelin Böhmer**
Department of Intelligent Systems
Delft University of Technology
2628 CD Delft, The Netherlands
`j.w.bohmer@tudelft.nl`

**Matthijs T. J. Spaan**
Department of Intelligent Systems
Delft University of Technology
2628 CD Delft, The Netherlands
`m.t.j.spaan@tudelft.nl`

## Abstract

To learn approximately optimal acting policies for decision problems, modern Actor Critic algorithms rely on deep Neural Networks (DNNs) to parameterize the acting policy and *greedification operators* to iteratively improve it. The reliance on DNNs suggests an improvement that is gradient based, which is per step much less greedy than the improvement possible by greedier operators such as the greedy update used by Q-learning algorithms. On the other hand, slow changes to the policy can also be beneficial for the stability of the learning process, resulting in a tradeoff between greedification and stability. To better address this tradeoff, we propose to decouple the acting policy from the policy evaluated by the critic. This allows the agent to separately improve the critic's policy (e.g. *value improvement*) with greedier updates while maintaining the slow gradient-based improvement to the parameterized acting policy. We investigate the convergence of this approach in the finite-horizon domain using a popular analysis scheme which generalizes Policy Iteration with arbitrary improvement operators and approximate evaluation. Empirically, incorporating value-improvement into the popular off-policy actor-critic algorithms TD3 and SAC significantly improves or matches performance over the baselines respectively, across different environments from the DeepMind continuous control domain, with negligible compute and implementation cost[1].

## 1 Introduction

The objective of Reinforcement Learning (RL) is to learn acting policies $\pi$, a probability distribution over actions, that, when executed, maximize the expected return (i.e., value) in a given task. Modern RL methods of the Actor-Critic (AC) family (e.g., Schulman et al., 2017; Fujimoto et al., 2018; Haarnoja et al., 2018b; Abdolmaleki et al., 2018) use deep neural networks to parameterize the acting

---

[1]Code is available at https://github.com/YanivO1123/viac.

policy, which is iteratively improved using variations of *policy improvement operators* based in stochastic gradient-descent (SGD), e.g., the policy gradient (Sutton et al., 1999). These methods rely on a specific type of policy improvement operators called *greedification operators*, which produce a new policy $\pi'$ that increases the current evaluation $Q^\pi$ (see Definition 2 for more detail).

In gradient-based optimization the magnitude of the update to the policy - the amount of greedification - is governed by the learning rate, which cannot be tuned independently to induce the maximum greedification possible at every step, the greedy update $\pi(s) = \arg\max_a Q^\pi(s, a)$ (which we define generally as any policy $\pi$ that has support only on maximizing actions). Similarly, executing $N$ repeating gradient steps with respect to the same batch will encourage the parameters to over-fit to the batch (as well as being computationally intensive) and is thus does not address the problem of limited greedification of gradient based operators. For these reasons, the greedification of DNN-based policies is typically slow compared to, for instance, the $\arg\max$ greedification used in Policy Iteration (Sutton & Barto, 2018) and Q-learning (Mnih et al., 2015).

While limited greedification can slow down learning, previous work has shown that too much greedification can cause instability in the learning process through overestimation bias (see van Hasselt et al., 2016; Böhmer et al., 2016; Fujimoto et al., 2018), which can be addressed through softer, less-greedy updates (Fox et al., 2016). This leads to a direct tradeoff between *greedification* and *learning stability*.

Previous work partially addresses this tradeoff by decoupling the policy improvement into two steps. First, an improved policy with *controllable* greediness is explicitly produced by a greedification operator as a target. Second, the acting policy is regressed against this target using supervised learning loss, such as cross-entropy. The target policy is usually not a DNN, and can be for instance a Monte Carlo Tree Search-based policy, a variational parametric distribution, or a nonparametric model (see Haarnoja et al., 2018b; Abdolmaleki et al., 2018; Grill et al., 2020; Hessel et al., 2021; Danihelka et al., 2022). Unfortunately, this approach does not address the tradeoff fully: The parameterized acting policy is still improved with gradient-based optimization which imposes similar limitations on the rate of change to the acting policy.

To better address this tradeoff, we propose to explicitly decouple the acting policy from the *evaluated* policy (the policy evaluated by the critic), and apply greedification independently to both. This allows for (i) the evaluation of policies that need not be parameterized and can be arbitrarily greedy, while (ii) maintaining the slower policy improvement to the acting policy that is suitable for DNNs and facilitates learning stability. We refer to an update step which evaluates an independently-improved policy as a *value improvement* step and to this approach as Value-Improved Actor Critic (VIAC). Since this framework diverges from the assumption made by the majority of RL methods (evaluated policy $\equiv$ acting policy) it is unclear whether this approach converges and for which improvement operators. Our first result is that *policy improvement* is not a sufficient condition for convergence to the optimal policy of even exact Policy Iteration algorithms because it allows for infinitesimal improvement. To classify improvement operators that guarantee convergence, we identify necessary and sufficient conditions for operators to guarantee convergence to an optimal policy for a family of generalized Approximate Policy Iteration algorithms, a popular setup for underlying-convergence analysis of AC algorithms (Tsitsiklis, 2002; Smirnova & Dohmatob, 2019).

We prove convergence for this class of operators in both generalized Approximate Policy Iteration and Value-Improved generalized Approximate Policy Iteration algorithms in finite-horizon MDPs. Prior work has shown that the generalized Approximate Policy Iteration setup converges for specific operators, as well as for all operators that induce deterministic policies (see Williams & Baird III, 1993; Tsitsiklis, 2002; Bertsekas, 2011; Smirnova & Dohmatob, 2019). Our results complement prior work by extending convergence to stochastic policies and a large class of practical operators, such as the operator developed for the Gumbel MuZero algorithm (Danihelka et al., 2022), as well as the Value-Improved extension to the algorithm. We demonstrate that incorporating value-improvement into practical algorithms can be beneficial with experiments in Deep Mind's control suite (Tunyasuvunakool et al., 2020) with the popular off-policy AC algorithms TD3 (Fujimoto et al., 2018) and SAC (Haarnoja et al., 2018b), where in all environments tested VI-TD3/SAC significantly outperform or match their respective baselines.

## 2 Background

The reinforcement learning problem is formulated as an agent interacting with a Markov Decision Process (MDP) $\mathcal{M}(\mathcal{S}, \mathcal{A}, P, R, \rho, H)$, where $\mathcal{S}$ a state space, $\mathcal{A}$ an action space, $P : \mathcal{S} \times \mathcal{A} \to \mathscr{P}(\mathcal{S})$

is a conditional probability measure over the state space that defines the transition probability $P(s, a)$. The immediate reward $R(s, a)$ is a state-action dependent bounded random variable. Initial states are sampled from the start-state distribution $\rho$. In finite horizon MDPs, $H$ specifies the length of a trajectory in the environment. Many RL setups and algorithms consider the infinite horizion case, where $H \to \infty$, but for simplicity's sake our theoretical analysis in Section 3 remains restricted to finite horizons, discrete state spaces and finite (and thus discrete) action spaces $|\mathcal{A}| < \infty$. Note that in finite horizon MDP the policy is not stationary, as the same state can have different optimal actions at different timesteps $t$ of an episode. We model this without loss of generality still as a stationary policy problem by augmenting the state with the decision time $t$, which casts an underlying state that is visited twice in an episode as two different states, which preserves stationarity of the policy in the augmented state space. The resulting state and transitions of the MDP then become a directed acyclic graph (DAG). Throughout the paper, we will assume all states in the MDP contain the timestep as part of the representation.

The objective of the agent is to find a policy $\pi : \mathcal{S} \to \mathscr{P}(\mathcal{A})$, a distribution over actions at each state, that maximizes the objective $J$, the expected return from the starting state distribution $\rho$. We denote the set of all possible policies with $\Pi$. This quantity can also be written as the expected state value $V^\pi$ with respect to starting states $s_0$:

$$J(\pi) = \mathbb{E}\big[V^\pi(s_0) \,\big|\, s_0 \sim \rho\big] = \mathbb{E}\Big[\sum_{t=0}^{H-1} \gamma^t r_t \,\Big|\, {}^{s_0 \sim \rho,\, s_{t+1} \sim P(s_t, a_t)}_{a_t \sim \pi(s_t),\, r_t \sim R(s_t, a_t)}\Big].$$

The discount factor $0 < \gamma \leq 1$ is traditionally set to 1 in finite horizon MDPs. The state value $V^\pi$ can also be used to define a state-action $Q$-value and vice versa, i.e., $\forall s \in \mathcal{S}, \forall a \in \mathcal{A}$:

$$Q^\pi(s, a) = \mathbb{E}\Big[r + \gamma V^\pi(s') \,\Big|\, {}^{r \sim R(s,a)}_{s' \sim P(s,a)}\Big], \qquad V^\pi(s) = \mathbb{E}\big[Q^\pi(s, a) \,\big|\, a \sim \pi(s)\big].$$

We refer to the optimal policy $\pi^* = \arg\max_\pi V^\pi$ and its value as $V^*$ and $Q^*$ respectively.

**Policy Improvement** To find $\pi^*$, many RL and Dynamic Programming (DP) approaches based in approximate or exact Policy Iteration (Sutton & Barto, 2018) can be cast as iterative processes that aim to produce a sequence of policies $\pi_n$ that *improve* over iterations such that the exact values $V^{\pi_n}$ or approximate values $v^{\pi_n} \approx V^{\pi_n}$ satisfy $V^{\pi_{n+1}} > V^{\pi_n}$ (or in the approximate case $v^{\pi_{n+1}} \gtrapprox v^{\pi_n}$) using *policy improvement operators*:

**Definition 1** (Policy Improvement Operator). *If an operator $\mathcal{I} : \Pi \to \Pi$ satisfies:*

$$\forall \pi \in \Pi, \forall s \in S: \; V^{\mathcal{I}(\pi)}(s) \geq V^\pi(s), \quad \forall \pi \in \Pi, \exists s \in S: \; V^{\mathcal{I}(\pi)}(s) > V^\pi(s) \tag{1}$$

*(i.e., policy improvement), as long as $\pi$ is not yet an optimal policy $V^\pi \neq V^*$, we call $\mathcal{I}$ a policy improvement operator.*

**Greedification** The policy improvement theorem (Sutton & Barto, 2018) is a fundamental result in RL and DP theory, which connects the policy improvement optimization process to a specific maximization problem referred to in literature as *greedification* (see Chan et al., 2022). Greedification is the process of finding a policy $\pi'$ which increases another policy, $\pi$'s, evaluation $Q^\pi$ (Equation 2):

**Theorem 1** (Policy Improvement). *Let $\pi$ and $\pi'$ be two policies such that $\forall s \in \mathcal{S}$:*

$$\sum_{a \in \mathcal{A}} Q^\pi(s, a)\pi'(a|s) \geq \sum_{a \in \mathcal{A}} Q^\pi(s, a)\pi(a|s) := V^\pi(s). \tag{2}$$

$$\text{Then:} \quad V^{\pi'}(s) \geq V^\pi(s). \tag{3}$$

*In addition, if there is strict inequality of Equation 2 at any state, then there must be strict inequality of Equation 3 at at least one state.*

We refer to Sutton & Barto (2018) for proof. Theorem 1 proves that when the evaluation $Q^\pi$ is exact, greedification with respect to $\pi, Q^\pi$ produces an improved policy $\pi'$. If the inequality in Equation 2 is strict $>$ we call $\pi'$ greedier than $\pi$ and any policy $\pi'$ such that $\sum_{a \in \mathcal{A}} Q^\pi(s, a)\pi'(a|s) = \max_{a \in \mathcal{A}} Q^\pi(s, a)$ a greedy policy with respect to $Q^\pi$.

**Greedification Operators** The policy improvement theorem and greedification give rise to the most popular class of policy improvement operators, *greedification operators*, which produce policy improvement (Equation 3) specifically by greedification:

**Definition 2** (Greedification Operator). *If an operator $\mathcal{I} : \Pi \times \mathcal{Q} \to \Pi$ satisfies:*

$$\sum_{a \in \mathcal{A}} \mathcal{I}(\pi, q)(a|s)q(s,a) \geq \sum_{a \in \mathcal{A}} \pi(a|s)q(s,a), \quad \forall \pi \in \Pi, \forall q \in \mathcal{Q}, \forall s \in S, \tag{4}$$

*as well as $\exists s \in \mathcal{S}$ such that:*

$$\sum_{a \in \mathcal{A}} \mathcal{I}(\pi, q)(a|s)q(s,a) > \sum_{a \in \mathcal{A}} \pi(a|s)q(s,a), \quad \forall \pi \in \Pi, \forall q \in \mathcal{Q}, \tag{5}$$

*unless $\pi$ is already greedy with respect to $q$: $\sum_{a \in \mathcal{A}} \pi(a|s)q(s,a) = \max_a q(s,a), \forall s \in \mathcal{S}$, we call $\mathcal{I}$ a greedification operator.*

The set $\mathcal{Q}$ denotes all bounded functions $q : \mathcal{S} \times \mathcal{A} \to \mathbb{R}$. Since practical operators are not generally designed to distinguish between exact $Q^\pi$ and approximated $q \approx Q^\pi$, we formulate the definition more generally in terms of $q \in \mathcal{Q}$. Greedification operators are policy improvement operators for $q = Q^\pi$ (i.e., Theorem 1). Although most of the analysis in this paper will focus on greedification operators, the problem we point to in our first theoretical result is not unique to greedification and applies to policy improvement operators in general, which motivates us to explicitly distinguish between the two. We provide an example of a policy improvement operator that is not a greedification operator in Appendix A.2, to demonstrate that greedification operators are a strict subset of policy improvement operators (when $q = Q^\pi$).

Perhaps the most famous greedification operator is *the greedy operator* $\mathcal{I}_{\arg\max}(\pi, q)(s) = \arg\max_a q(s,a)$, which drives foundational algorithms such as Value Iteration, Policy Iteration and Q-learning (Sutton & Barto, 2018). Many modern RL methods on the other hand are based in the actor critic (AC) framework, which we generally refer to as the iteration of (approximate) *policy improvement* (improving the actor), (approximate) *policy evaluation* (evaluating the actor, i.e., updating the critic). ACs traditionally rely on variations of the *policy gradient* operator (Sutton et al., 1999), which is well suited for the greedification of parameterized policies.

Other popular greedification operators are *deterministic greedification* operators $\mathcal{I}_{det}$ (Williams & Baird III, 1993) which produce policies that are greedier (Equation 2) and deterministic. The regularized-policy improvement operator (see Grill et al., 2020) used by Gumbel MuZero (Danihelka et al., 2022) $\mathcal{I}_{gmz}(\pi, q)(s) = \text{softmax}(\sigma(q(s,\cdot)) + \log \pi(s))$ (for $\sigma$ a monotonically increasing transformation). Best-of-N (BoN), a popular operator in large language model alignment with RL (Gui et al., 2024; Huang et al., 2025). At a state $s$ BoN samples $N$ actions $A_N = \{a_1, \ldots, a_N\}$ from $\pi(s)$ and evaluates them using $q(s, a_i)$. BON returns the maximizing action: $\mathcal{I}_{BON}(\pi, q)(s) = \arg\max_{a \in A_N} q(s,a)$. As the sample size $N$ increases, BoN better approximates the $\arg\max_{a \in \mathcal{A}} q(s,a)$.

**Implicit greedification operators** Recently, Kostrikov et al. (2022) proposed that it is also possible to produce *implicit* greedification, by training a critic to approximate the value of a greedier policy directly, without that policy being explicitly defined. The authors demonstrate that by training a critic $v_\psi$ with the asymmetric expectile loss $\mathcal{L}_2^\tau$ on a data set $\mathcal{D}$ drawn with policy $\pi$,

$$\mathcal{L}(\theta) = \mathbb{E}\left[\mathcal{L}_2^\tau\left(v_\psi(s), Q^\pi(s,a)\right)\big| s, a \sim \mathcal{D}\right], \quad \mathcal{L}_2^\tau(x,y) = |\tau - \mathbb{1}_{y-x<0}|(y-x)^2, \tag{6}$$

for $\tau > \frac{1}{2}$ the critic $v_\psi(s)$ directly estimates the value of a policy than is greedier than $\pi$, with $\tau \to 1$ corresponding to the value of an $\arg\max$ policy. This operator is then used to drive their Implicit Q-learning (IQL) algorithm for offline-RL, where the $\mathcal{L}_2^\tau$ enables the critic to approximate the value of an *optimal* policy without the bootstrapping of actions that are out of the training distribution.

**Generalized Policy Iteration** A popular approach for analyzing the underlying convergence behavior of large families of RL algorithms is to analyze the convergence of a DP algorithm which abstracts the underlying learning dynamics of the RL algorithms (for example, see Smirnova & Dohmatob, 2019). A common setup used for this analysis is that of an approximate Policy Iteration algorithm (sometimes called specifically Optimistic or Modified Policy Iteration, see (Bertsekas, 2011)) which iterates *policy improvement*, *approximate policy evaluation*. The policy improvement step most often uses the greedy operator. The policy evaluation step generalizes across Value Iteration and Policy Iteration, by doing a finite number $k$ of Bellman updates with the same policy $q_{i+1}(s,a) = \mathcal{T}^\pi q_i(s,a) = \mathbb{E}[R(s,a)] + \gamma \mathbb{E}_{s' \sim P}[\sum_{a' \in \mathcal{A}} \pi(a'|s')q_i(s',a')], i = 1, \ldots, k$. Since our objective in this work is to analyze the underlying convergence of RL algorithms with respect to as many possible policy improvement operators, we formulate a variation of this algorithm which is generalized to all policy improvement operators $\mathcal{I}$ (Algorithm 1), which we refer to as Generalized Policy Iteration (GPI). $\mathcal{T}^*$ denotes the Bellman optimality operator, $\mathcal{T}^* q_i(s,a) = \mathbb{E}[R(s,a)] + \gamma \mathbb{E}_{s' \sim P}[\max_{a' \in \mathcal{A}} q(s',a')]$.

---

**Algorithm 1** Generalized Policy Iteration

---
1: For starting functions $q \in \mathcal{Q}$, $\pi \in \Pi$ greedification operator $\mathcal{I}$, $k \geq 1$ and $\epsilon > 0$
2: **while** $|\sum_{a \in \mathcal{A}} (\pi(a|s)q(s,a)) - \max_b q(s,b)| > 0, \forall s \in \mathcal{S}$ and $|q(s,a) - \mathcal{T}^* q(s,a)| > 0$ **do**
3:      $q(s,a) \leftarrow (\mathcal{T}^\pi)^k q(s,a), \forall (s,a) \in \mathcal{S} \times \mathcal{A}$
4:      $\pi(s) \leftarrow \mathcal{I}(\pi, q)(s), \forall s \in \mathcal{S}$

---

## 3 Value Improved Generalized Policy Iteration Algorithms

Since the framework of Value-Improved AC/GPI generalizes AC/GPI beyond algorithms that evaluate their own policy, our first objective is to analyze the the underlying convergence of this family of algorithms under a very general setup. We begin by extending the DP framework of GPI, which underlies ACs, to a DP framework which underlies VIACs. The framework is extended by decoupling the improvement of the *acting* policy from that of the *evaluated* policy (line 3 in Algorithm 1). We refer to this new framework as Value-Improved GPI (Algorithm 2). Modifications to the original algorithm are marked in blue. Since the acting and evaluated policies are improved with different operators $\mathcal{I}_1$ and $\mathcal{I}_2$, it is not apparent whether $\pi$ of Algorithm 2 converges to the optimal policy, i.e. whether decoupling the policies is sound. Therefore, our aim is to establish general pairs of operators for which this process converges.

---

**Algorithm 2** Value-Improved Generalized Policy Iteration

---
1: For starting vectors $q \in \mathcal{Q}$, $\pi \in \Pi$, policy improvement operators $\mathcal{I}_1, \mathcal{I}_2$, $k \geq 1$
2: **while** $|\sum_{a \in \mathcal{A}} (\pi(a|s)q(s,a)) - \max_b q(s,b)| > 0, \forall s \in \mathcal{S}$ and $|q(s,a) - \mathcal{T}^* q(s,a)| > 0$ **do**
3:      $q(s,a) \leftarrow (\mathcal{T}^{\mathcal{I}_2(\pi,q)})^k q, \forall (s,a) \in \mathcal{S} \times \mathcal{A}$
4:      $\pi(s) \leftarrow \mathcal{I}_1(\pi, q)(s), \forall s \in \mathcal{S}$

---

A fundamental result in RL is that policy iteration algorithms converge to the optimal policy for any policy improvement operator (Definition 1 and by extension, all greedification operators) that produces deterministic policies. This holds because a finite MDP has only a finite number of deterministic policies through which the policy iteration process iterates (Sutton & Barto, 2018). This result however does not generalize to operators that produce *stochastic* policies, which are used by many practical RL algorithms such as PPO (Schulman et al., 2017), MPO (Abdolmaleki et al., 2018), SAC (Haarnoja et al., 2018b), and Gumbel MuZero (Danihelka et al., 2022). Our first theoretical result is that for stochastic policies, the policy improvement property (whether satisfied through greedification or not) is not sufficient to guarantee convergence, even in the limiting case of Policy Iteration with exact evaluation.

**Theorem 2** (Improvement is not enough). *Policy improvement is not a sufficient condition for the convergence of Policy Iteration algorithms (Algorithm 1 with exact evaluation) to the optimal policy for all starting policies $\pi_0 \in \Pi$ in all finite-state MDPs.*

**Proof sketch.** With stochastic policies, an infinitesimal policy improvement is possible, which can satisfy the policy improvement condition at every step and yet converge in the limit to policies that are *not* $\arg\max$ policies. Since every optimal policy is an $\arg\max$ policy (note that we define $\arg\max$ policies as policies with support only on maximizing actions, not necessarily as deterministic policies), Policy Iteration with such operators cannot be guaranteed to converge to the optimal policy. For a complete proof see Appendix A.1.

**Why is this a problem?** Many algorithms (for example, GumbelMZ Danihelka et al. (2022)) are motivated by policy improvement through demonstrating greedification. Theorem 2 demonstrates that this is not sufficient to establish that the resulting policy improvement will lead to an optimal policy. For that reason, convergence for these algorithms must generally proven individually for each new operator (e.g., see MPO and GreedyAC Chan et al. (2022)), which is often an arduous and nontrivial process.

Furthermore, Theorem 2 and its underlying intuition highlight a critical gap: we currently lack guiding principles for designing novel greedification operators in the form of necessary and sufficient conditions for convergence to the optimal policy. To illustrate that this can lead to problems in practice, we show in Appendices A.5 and A.6 that choices of the $\sigma$ used by the greedification operator

$\mathcal{I}_{gmz}$ can render this operator either sufficient or insufficient. To address this problem, we identify a necessary condition and two independent sufficient conditions for greedification operators, such that they induce convergence of Algorithm 1.

**Definition 3** (Necessary Greedification). *In the limit of $n$ applications of a greedification operator $\mathcal{I}$ on a value estimate $q \in \mathcal{Q}$ and a starting policy $\pi_0 \in \Pi$, the policy $\pi_n$ converges to a greedy policy with respect to $q$, $\forall s \in \mathcal{S}$:*

$$\lim_{n \to \infty} \sum_{a \in \mathcal{A}} q(s, a) \pi_n(a|s) = \max_a q(s, a), \quad where \quad \pi_{n+1}(s) = \mathcal{I}(\pi_n, q)(s). \tag{7}$$

**Intuition.** Since every optimal policy is an $\arg\max$ policy (has support only on actions that maximize $Q^*$), if a greedification operator cannot converge to an $\arg\max$ policy even in the limit and for a fixed $q$ then this operator cannot converge to an optimal policy in general. See Appendix A.1 for a concrete example where such a condition is necessary for convergence of a Policy Iteration algorithm. It is possible to formulate the same condition more specifically for the set of all $Q$ functions $\{Q^\pi \mid \forall \pi \in \Pi\}$. However, since we are interested in algorithms that may not have access to exact values $Q^\pi$, it seems more prudent to define it more generally $\forall q \in \mathcal{Q}$. Since practical operators are not generally designed to distinguish between exact $Q^\pi$ and approximated $q \approx Q^\pi$, we formulate the definition more generally in terms of $q \in \mathcal{Q}$.

Unfortunately, the necessary greedification condition is not sufficient, even in the case of exact evaluation. This is due to the fact that assuming convergence to a greedy policy in the limit for a *fixed* $q$ function does not necessarily imply the same when the $q$ function changes between iterations. There exist settings where the ordering of actions $a, a', q(s, a) < q(s, a')$ can oscillate between iterations, preventing the convergence to greedy policies (See Appendix A.3 for a concrete example). Below, we identify two additional conditions which are each *sufficient* for convergence in finite horizon and finite action spaces. The first condition resolves the issue by lower-bounding the rate of improvement, which guarantees that the oscillation does not continue infinitely. The second simply augments the necessary greedification condition to require convergence for *any* sequence of Q functions.

**Definition 4** (Lower Bounded Greedification). *We call an operator $\mathcal{I}$ a lower-bounded greedification operator if $\mathcal{I}$ is a greedification operator (Definition 2) and for every $q \in \mathcal{Q}$, $\exists \epsilon > 0$, such that $\forall s \in \mathcal{S}$ and $\forall \pi \in \Pi$:*

$$\sum_{a \in \mathcal{A}} \mathcal{I}(\pi, q)(a|s) q(s, a) - \sum_{a \in \mathcal{A}} \pi(a|s) q(s, a) \quad > \quad \epsilon,$$

*unless $\sum_{a \in \mathcal{A}} \mathcal{I}(\pi, q)(a|s) q(s, a) = \max_a q(s, a)$, $\forall s \in \mathcal{S}$.*

**Intuition.** Since the lower bound $\epsilon$ is constant with respect to a stationary $q$, it eliminates the possibility of infinitesimal improvements and guarantees convergence to an $\arg\max$ policy in finite iterations with respect to the stationary $q$ (See Lemma 8 and Appendix A.12 for proof). We note that this definition does not guarantee convergence to an optimal policy nor an $\arg\max$ policy with respect to a *non-stationary* $q_n$.

**Definition 5** (Limit-Sufficient Greedification). *Let $q_0, q_1, \dots \in \mathcal{Q}$ be a sequence of functions such that $\lim_{n \to \infty} q_n = q$ for some $q \in \mathcal{Q}$. Let $\pi_0, \pi_1, \dots$ be a sequence of policies where $\pi_{n+1} = \mathcal{I}(\pi_n, q_{n+1})$ for some operator $\mathcal{I}$. We call an operator $\mathcal{I}$ a Sufficient greedification operator if $\mathcal{I}$ is a greedification operator (Definition 2) and in the limit $n \to \infty$ the improved policy $\pi_{n+1}$ converges to a greedy policy with respect to the limiting value $q$, $\forall s \in \mathcal{S}$:*

$$\lim_{n \to \infty} \sum_{a \in \mathcal{A}} \pi_n(a|s) q_n(s, a) = \max_a q(s, a). \tag{8}$$

**Intuition.** Even in the presence of infinitesimal improvement and non-stationary estimates $q_n$, a limit sufficient greedification operator is guaranteed to converge to a greedy policy in the limit $n \to \infty$, as long as there exists a limiting value on the sequence of value estimates $\lim_{n \to \infty} q_n = q$.

**Practical operators that are sufficient operators** Lower bounded greedification is used to establish convergence for MPO (see Appendix A.2, Proposition 3 of (Abdolmaleki et al., 2018)). Similarly, deterministic operators $\mathcal{I}_{det}$ are also lower bounded greedification operators (see Appendix A.4 for proof). Lower bounded greedification operators however cannot contain operators that induce convergence to the greedy policy *only* in the limit, because the convergence they induce is in finite steps. $\mathcal{I}_{gmz}$ on the other hand induces convergence only in the limit, and in fact is more generally a

limit-sufficient greedification operator (see Appendix A.5 for proof). The deterministic greedification operator on the other hand does not converge with respect to arbitrary non-stationary sequences $\lim_{n\to\infty} q_n$ (see Appendix A.7), which leads us to conclude that both sets are useful in that they both contain practical operators and neither set contain the other. The greedy operator on the other hand is a member of *both* sets, demonstrating that the sets are also not disjoint (see Appendix A.8 for proof).

Equipped with Definitions 4 and 5 we establish our main theoretical result, convergence for both Algorithms 1 and 2 for all sufficient greedification operators:

**Theorem 3** (Convergence of Algorithms 1 and 2). *Generalized Policy Iteration algorithms and their Value Improved extension (Algorithms 1 and 2 respectively) converge for sufficient greedification operators, in finite iterations (for operators defined in Definition 4) or in the limit (for operators defined in Definition 5), in finite-horizon MDPs.*

**Proof sketch:** Using induction from terminal states, the proof builds on the immediate convergence of values of terminal states $s_H$, convergence of policies at states $s_{H-1}$ and finally on showing that given that $q, \pi$ converge for all states $s_{t+1}$, they also converge for all states $s_t$ (in finite iterations or in the limit, respectively). The evaluation of a *greedier* policy (value improvement, line 3 in Algorithm 2) is accepted by the induction that underlies the convergence of Algorithm 1 which allows us to use the same method to establish convergence for Algorithm 2. The full proof is provided in the Appendix. In A.9 for Algorithm 1 with limit sufficient operators and $k = 1$, extended to $k \geq 1$ in A.10, to Value-Improved algorithms in A.11, and to lower-bounded operators in A.12.

A corollary of $\mathcal{I}_{gmz}$ being a limit-sufficient greedification operator along with Theorem 3 is the convergence of a process underlying the Gumbel MuZero algorithm.

**Corollary 1.** *The Generalized Policy Iteration process underlying the Gumbel MuZero algorithm family converges to the optimal policy for finite horizon MDPs, for all $\pi_0 \in \Pi$ such that $\log \pi(a|s)$ is defined $\forall s \in \mathcal{S}, a \in \mathcal{A}$.*

Interestingly, $\mathcal{I}_2$ does not need to be a sufficient or even a necessary greedification operator for convergence to the optimal policy, as is established by the following corollary:

**Corollary 2.** *Algorithm 2 converges to the optimal policy for any non-detriment operator $\mathcal{I}_2$ (e.g., operators that satisfy the non-strict inequality of Equation 4), as long as $\mathcal{I}_1$ is itself sufficient.*

For proof see Appendix A.11. As another consequence, Corollary 2 also establishes that the algorithm converges when implicit greedification operators such as the expectile loss are used for value improvement. Motivated that the Generalized Policy Iteration process underlying VIAC algorithms converges, we proceed to empirically evaluate practical RL VIAC algorithms.

## 4 Value Improved Actor Critic Algorithms

Value-improvement can be incorporated into existing AC algorithms in one of two ways: (i) Incorporating an additional *explicit* greedification operator to produce a greedier evaluation policy, and use the greedier policy to bootstrap actions from which to generate value targets. (ii) Incorporating an *implicit* greedification operator, for example by replacing the value loss with an asymmetric loss (Algorithms 3 and 4 respectively in Appendix C and implementation details in Appendix D). We begin by testing the hypothesis that additional greedification of the evaluation policy (e.g., value improvement) can directly lead to accelerated learning by extending TD3 (Fujimoto et al., 2018) with value improvement (VI-TD3, Algorithm 3).

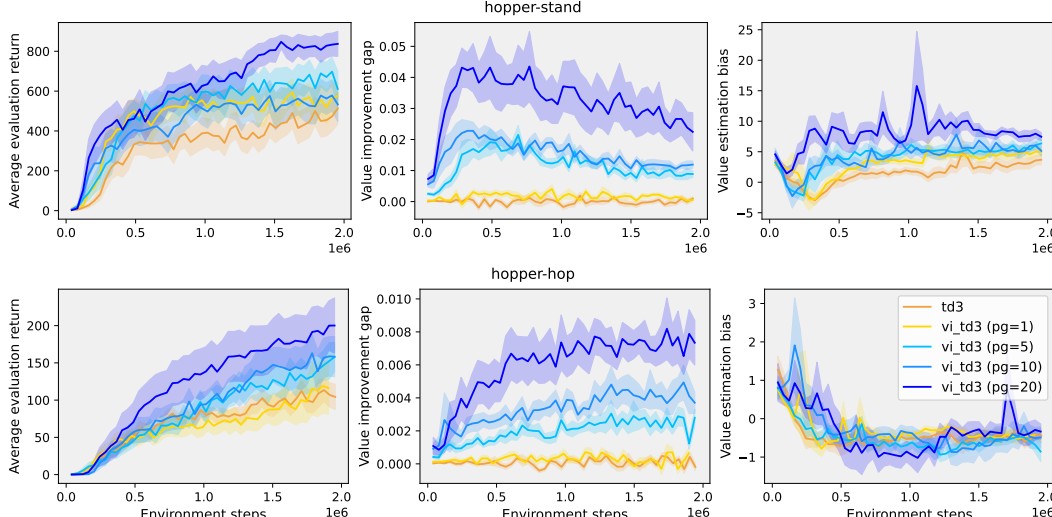

Figure 1: Mean and one standard error in the shaded area across 10 seeds for VI-TD3 with $\mathcal{I}_2$ the deterministic policy gradient and increasing number of gradient steps (pg=n), with baseline (pg=0) TD3 for reference.

We choose the same improvement operator already used by TD3 $\mathcal{I}_2 = \mathcal{I}_1$, the deterministic policy gradient, as a value improvement operator, in order to decouple possible effects stemming from the combination of different operators. In order to compare different degrees of greedification, a different number $pg = n$ of repeating gradient steps with respect to the same batch are applied to the *evaluated* policy, which is then discarded after each use. We evaluate the performance of the agents in classic control environments from the DeepMind continuous control benchmark (Tunyasuvunakool et al., 2020). Results are presented in Figure 1, where performance increases with greedification (left).

As expected, greedier targets are larger targets (center), that is the difference between the value bootstrap that uses the greedier policy $\pi'$ and the baseline bootstrap increases with greedification. An interaction with over-estimation bias (which we compute in the same manner as Chen et al. (2021)) exists in some environments like hopper-stand, but cannot explain the improved performance exclusively, as shown in the hopper-hop environment (Figure 1, right).

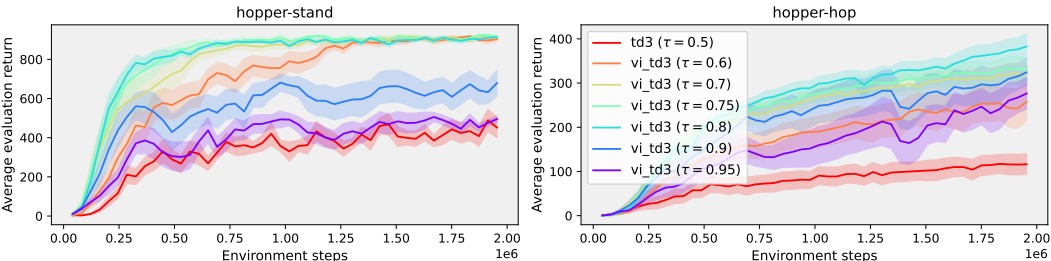

Figure 2: Mean and one standard error across 10 seeds for VI-TD3 with expectile loss with different values of the expectile parameter $\tau$. Performance increases up to $\tau = 0.8$ and then decays.

Repeating gradient steps are very computationally expensive however. Implicit policy improvement on the other hand provides value improvement for negligible compute and implementation cost. In addition, the greedification amount can be chosen with the greedification parameter $\tau$ directly. In Figure 2 we evaluate VI-TD3 with implicit value improvement (Algorithm 4) based in the expectile loss operator and increasing values of the greedification-parameter $\tau$. The increased greedification monotonically improves performance up to a point, from which performance monotonically degrades, suggesting that the greedification of the evaluated policy can be tuned as a hyperparameter. This supports the conclusions of previous literature that there is a tradeoff between stability and greedification, and suggests that this tradeoff can be at least partially addressed with value improvement.

In Figure 3 we compare baseline TD3 and SAC (Haarnoja et al., 2018b) to VI-TD3 and VI-SAC with implicit value improvement ($\tau = 0.75$, blue) across a larger number of control environments.

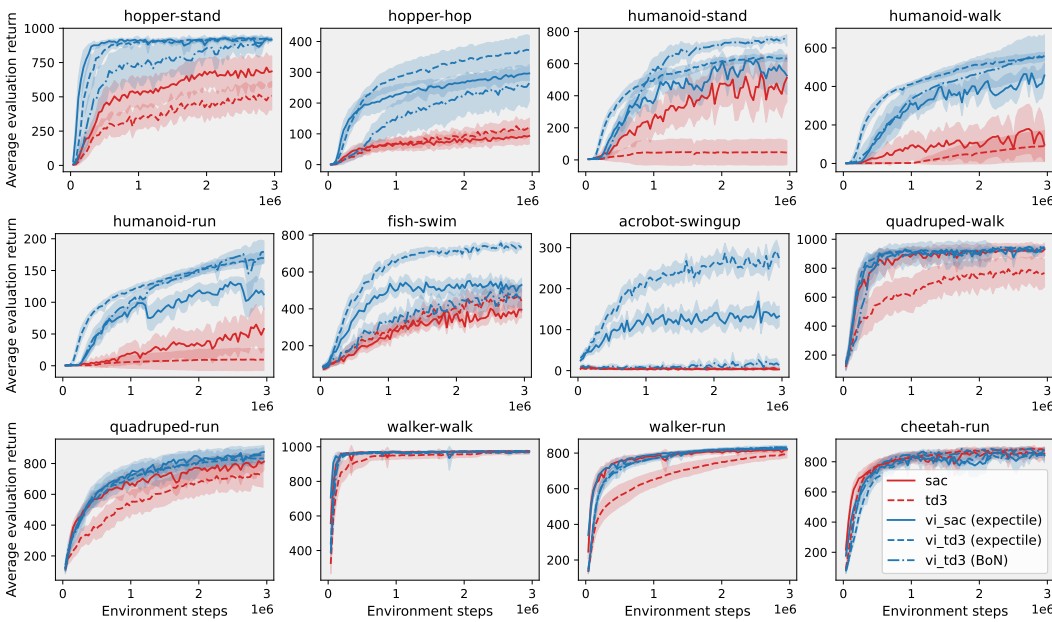

Figure 3: Baseline TD3 (dashed) and SAC (solid) in red vs. VI-TD3/SAC (blue) with expectile loss and BoN (dot-dashed). Mean and two standard errors ($\approx 95\%$ Gaussian CI) in the shaded area of evaluation curves across 10 seeds for BoN and 20 for the other agents.

Across all environments, the VI (blue) variations significantly outperform or matches their respective baseline (red) while introducing negligible additional compute and implementation cost.

The significant performance increase provided by the expectile operator in Figure 3 raises the possibility that the performance gains are not due to the greedification of the evaluated policy, however. Rather, the implicit expectile operator itself may be responsible for the performance gain. To investigate whether this is the case, we include an additional VI-TD3 variation with an explicit improvement operator operator: Best-of-N (BoN, dot-dashed, see Section 2). BoN-based VI-TD3 ($N = 64$) provides similar performance gains to expectile-based VI-TD3 in most environments, demonstrating that significant performance gains in these domains are not unique to the expectile operator.

Along similar lines, although Figure 1 demonstrates that it is possible to see performance gains that are decoupled from overestimation increase, that may not be the case for the expectile operator. It is possible that the (overall more significant) performance gains of the expectile operator are instead resulting from strongly-increased over-estimation bias. In Figure 4 in Appendix B we investigate in more detail the interaction between greedification, performance and over estimation bias per environment for the expectile operator. We find that in the majority of the environments significant performance gains are observed without any over estimation increase. We summarize these results in Table 1 in Appendix B.

We include additional results in Appendix B: (i) In Figure 5 we include results for the recent algorithm TD7 (Fujimoto et al., 2023), which shows similar performance gains with value improvement. (ii) In Figure 6 we investigate increasing the greediness of the update to the parameterized acting policy. Repeating gradient steps on the same batch do not effectively increase sample efficiency, as discussed in Section 1. (iii) In Figure 7 we investigate the tradeoff between spending the compute of additional gradient steps for value improvement vs. for increased replay ratio. In line with similar findings in literature (Chen et al., 2021), replay ratio provides a strong performance gain for small ratios. However, as the ratio increases, performance degrades, a result which the literature generally attributes to instability. The VI agent on the other hand does not degrade with the same increase to compute. (iv) In Figure 8 we investigate more generally how the greedification of the acting policy compares to the greedification of the evaluated policy - is one more important than the other? Can value improvement replace policy improvement? Our results suggest that as long as the Value-Improvement operator is rooted in a policy $\pi_i$ maintained during the learning process, the rate at which this policy is improved is the more impactful factor. On the other hand, slow policy improvement does not prevent value-improvement from significantly accelerating the learning process.

# 5 Related Work

VIAC generalizes the underlying learning scheme of several recent methods. XQL (Garg et al., 2023), SQL, and EQL (Xu et al., 2023) build on the implicit policy improvement setup of IQL to design algorithms that iterate a *value improvement → policy extraction* loop, rather than the *policy improvement → policy evaluation* loop of standard AC methods. When the policy extraction step (learning a policy given a value function) can be interpreted as a policy improvement operator, these algorithms become members of the VIAC family with specific instantiations of operators. BEE (Ji et al., 2024) is another recent method that explicitly considers a value target that includes its own maximization operator, albeit from the perspective of mixing targets that take into account exploration / exploitation tradeoffs in the environment. Similarly, OBAC Luo et al. (2024) learns a pair of value functions, one for the acting policy and one using implicit policy improvement over data from the replay buffer and uses both pairs of value functions to train the acting policy.

In model-based RL, it is popular to employ improvement operators for action selection and policy improvement, and sometimes even to generate value targets (e.g., value improvement, for example see Moerland et al., 2023). The more common setup employs the same operator for action selection and policy improvement (for example, AlphaZero (Silver et al., 2018)). MuZero Reanalyze (Schrittwieser et al., 2021) is an example of an algorithm that considers using the same operator (tree search in this case) to produce value targets as well, and thus can be thought of as belonging to the setup of VIAC. These algorithms however are traditionally motivated from the perspective that the acting policy, target policy and evaluated policy all coincide as they are all produced by the same operator.

TD3 can be thought of as an example of an agent which acts, improves, and evaluates three different policies: The acting policy is improved using the *deterministic policy gradient*, during action selection the acting policy is modified with noise in order to induce exploration, and finally the evaluated policy is regularized with a differently-parameterized noise in order to improve learning stability. Although only one policy improvement operator is used, TD3 can be thought of as an algorithm which decouples the acting policy from the evaluated policy. GreedyAC (Neumann et al., 2023) inspired a family of algorithms (see Lingwei Zhu, 2025) which share similarities with the VIAC framework in that they explicitly maintain two different policies, although both policies are used for policy improvement, as opposed to value improvement.

# 6 Conclusions

In this work we have proposed to decouple the evaluated policy from the acting policy and apply policy improvement additionally to the evaluated policy (*value improvement*) in order to better control the tradeoff between greedification and stability in Actor Critic (AC) algorithms. We refer to this approach as Value-Improved AC (VIAC). VIAC provides a unified perspective on recent online and offline RL algorithms which combine different improvement operators (Garg et al., 2023; Xu et al., 2023; Ji et al., 2024; Xu et al., 2025). We've shown that policy improvement is not a sufficient condition for convergence of Dynamic Programming / RL algorithms with stochastic policies. We've identified necessary and sufficient conditions for convergence of such algorithms and demonstrated that Generalized Policy Iteration (Algorithm 1), which underlies the learning dynamics of ACs, converges to the optimal policy for such sufficient greedification operators in the finite horizon domain. With this groundwork laid, we've demonstrated that Value-Improved Generalized Policy Iteration (Algorithm 2), which respectively underlies VIACs, converges to the optimal policy in the finite horizon domain.

Empirically, VI-TD3 and VI-SAC significantly improve upon or match the performance of their respective baselines in all DeepMind control environments tested with negligible increase in compute and implementation costs. We hope that our work will act as motivation to design future AC and general Approximate Policy Iteration algorithms with multiple improvement operators in mind, as well as extend existing algorithms with value-improvement.

## Acknowledgments

We thank Joery de Vries and Caroline Horsch for timely assistance with compute. Mustafa Mert Çelikok was partially funded by the Hybrid Intelligence Center, a 10-year programme funded by the Dutch Ministry of Education, Culture and Science through the Netherlands Organisation for Scientific Research, https://hybridintelligence-centre.nl, grant number 024.004.022. The project has

received funding from the EU Horizon 2020 programme under grant number 964505 (Epistemic AI). The computational resources for the experiments were provided by the Delft High Performance Computing Centre (DHPC) and the Delft Artificial Intelligence Cluster (DAIC).

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

# A Proofs

## A.1 Theorem 2: policy improvement is not enough

Theorem 2 states: *Policy improvement is not a sufficient condition for the convergence of Policy Iteration algorithms (Algorithm 1 with exact evaluation) to the optimal policy for all starting policies $\pi_0 \in \Pi$ in all finite-state MDPs.*

*Proof sketch:* We will construct a simple MDP where the $Q^\pi$ values remain the same for all policies $\pi$, and show that even in this simple example, it is possible for a Policy Iteration algorithm to converge to non-optimal policies, with policy improvement operators that allow for stochastic policies. In addition, while adjacent to the narrative of this paper, we demonstrate that the same problem persists with *deterministic* policies in *continuous* action spaces in Appendix A.1.1.

*Proof.* Consider a very simple deterministic MDP with starting state $s_0$ and two actions $a_1, a_2$ that lead respectively to two terminal states $s_1, s_2$. The reward function $R(s_0, a_1) = 1$ and $R(s_0, a_2) = 2$ and transition function $P(s_1|s_0, a_1) = 1, P(s_2|s_0, a_2) = 1$ and zero otherwise. We have $Q^\pi(s_0, a_1) = 1$ and $Q^\pi(s_0, a_2) = 2$ for all policies $\pi \in \Pi$. The optimal policy is therefor $\pi^*(s_0) = a_2$, that is $\pi^*(a_1|s_0) = 0$ and $\pi^*(a_2|s_0) = 1$.

Consider the very simple policy improvement operator $\mathcal{I}_{inadequate}$. When $\sum_{a \in \mathcal{A}} \pi(a|s)Q^\pi(s, a) < \sum_{a \in \mathcal{A}} \text{softmax}(Q^\pi(s, \cdot))(a|s)Q^\pi(s, a)$, the operator is defined as follows: $\mathcal{I}_{inadequate}(\pi)(s) = \alpha\pi(s) + (1 - \alpha)\text{softmax}(Q^\pi(s, \cdot))(s)$. Otherwise, when the policy is already greedier than the softmax policy, the operator is defined as follows: $\mathcal{I}_{inadequate}(\pi)(s) = \arg\max_a Q^\pi(s, a)$.

In natural language: when the policy is less greedy than the softmax policy, the operator produces a mix between the current policy and the softmax policy. This is always greedier than the current policy, and thus will act as a greedification operator for policies such policies. When the policy is as greedy or greedier than the softmax, the operator produces directly a greedy policy.

The policy improvement theorem proves that this operator is a Policy Improvement operator, because for every policy $\pi$ it greedifies the policy with respect to $Q^\pi$ (that is, the policy's evaluation is strictly larger unless the policy is already greedy).

We now apply this operator in the Policy Iteration scheme to the simple example MDP specified above, with a starting uniform policy $\pi(a_1|s_0) = \pi(a_2|s_0)$. Since this operator produces a mixture of the current and softmax policy, in the limit it will converge to the softmax policy $\lim_{n\to\infty} \pi_n = \text{softmax}(Q(s, \cdot)) \neq \arg\max_a Q(s, a)$, i.e. to a sub-optimal policy, despite being a policy improvement operator. This proves that policy improvement operators cannot in general guarantee convergence to the optimal policy even in the limiting setting of Policy Iteration, and thus also not in more general setting such as Approximate Policy Iteration. $\square$

In this example, since $Q^\pi$ does not change across different iterations $n$ of a Policy Iteration algorithm, we can identify the following: For convergence to the optimal policy, it is necessary that a greedification operator $\pi_{n+1} = \mathcal{I}(\pi_n, q)$ will converge to a greedy policy, with respect to the same stationary $q = Q^\pi$:

$$\lim_{n\to\infty} \sum_{a \in \mathcal{A}} \pi_n(a|s)q(s, a) = \max_a q(s, a), \forall s \in \mathcal{S}.$$

More generally, this example shows that we can construct general operators $\mathcal{I}$ that produce arbitrary sequences of policies $\pi_{n+1} = \mathcal{I}(\pi_n), n \geq 0$ with values $V^{\pi_{n+1}}$ such that $V^{\pi_{n+1}}(s) \geq V^{\pi_n}(s), \forall s \in \mathcal{S}$, with a strict $>$ in at least one state. Since this is the only condition required by policy improvement, operators $\mathcal{I}$ are policy improvement operators. However, since the improvement need not be bounded it is possible to construct sequences that converge to arbitrary values in the interval $(V^{\pi_0}(s), V^*(s)], \forall s \in \mathcal{S}$. I.e., because the improvement property allows for infinitesimal improvements, it does not guarantee convergence to the optimal policy.

### A.1.1 Policy improvement in continuous action spaces

A similar problem applies to continuous action spaces. Imagine a similar MDP as above with a continuous action spaces $\mathcal{A} = (0, 1)$, reward function $R(s_0, a) = a, \forall a \in A$ and zero otherwise and every transition is terminal. Now imagine an operator that produces a deterministic action $\mathcal{I}(\pi, q)(s)$, such that $\int Q^\pi(s, a)\mathcal{I}(\pi, Q^\pi)(a|s)ds > \int Q^\pi(s, a)\pi(a|s)ds$ unless the policy is already optimal. $\mathcal{I}$ is an improvement operator and satisfies the greedification property.

Let us again choose $\pi_0$ the uniform policy across actions. At the first step, $\mathcal{I}$ can produce just-above the middle action $a_1 > 0.5$. At each step, $\mathcal{I}$ can produce a new action $\mathcal{I}(\pi_1, Q^*)(s_0) = \pi_2(s_0) = a_2 > a_1$. However, since the space $(0, 1)$ is the continuum and non-countable, there are more actions to select that any iterative process will ever have to go through. Therefor, even in the limit $n \to \inf$, the operator will never have to choose $\mathcal{I}(\pi_n, Q^*) = 1$.

## A.2 Policy Improvement operators that are not Greedification operators

**Lemma 1.** *There exist operators $\mathcal{I} : \Pi \times \mathcal{Q} \to \Pi$ that are Policy Improvement operators, and therefore fulfill Equation 3, but are not Greedification operators according to Definition 2.*

*Proof sketch:* We will prove that a random-search operator that mutates the policy $\pi$ randomly into a new policy $\pi'$, evaluates $\pi'$ and keeps it if $V^\pi > V^{\pi'}$, is not a greedification operator, even though it is a policy improvement operator by definition. We do this by constructing an MDP and choosing a specific initial policy $\pi_0$. Greedification with respect to the initial policy, at state $s_0$, results in $\pi_1(s_0) = a_1$. However, the optimal policy in this state actually chooses action $a_0$, because the optimal policy can take better actions in the future than policy $\pi_1$. Such an example proves that it is possible to construct policy improvement while violating greedification, demonstrating that the condition only goes one way: every greedification operator is policy improvement operator, not vice versa.

*Proof.* Consider the following finite-horizon MDP: State space $\mathcal{S} = \{s_1, \ldots, s_{10}\}$. Action space $\mathcal{A} = a_1, a_2, a_3$. States $\{s_5, \ldots, s_{10}\}$ are terminal states. Transition function: $f(s_1, a_1) = s_2$, $f(s_1, a_2) = s_3$, $f(s_1, a_3) = s_4$, $f(s_2, a_1) = s_5$, $f(s_2, a_2) = s_6$, $f(s_3, a_1) = s_7$, $f(s_3, a_2) = s_8$, $f(s_4, a_1) = s_9$, $f(s_4, a_2) = s_{10}$.
Rewards: $R(s_2, s_5) = 2$, $R(s_2, s_6) = -1$, $R(s_3, s_7) = 1$, $R(s_3, s_8) = 0$, $R(s_4, s_9) = 3$, $R(s_4, s_{10}) = -2$.
Actions that are not specified lead directly to a terminal state with zero reward.

Let us begin by identifying the optimal policy in this MDP, in states $s_1$ and $s_4$: $\pi^*(s_1) = a_3$ and $\pi^*(s_4) = a_1$, with a value of 3 without a discount factor.

Let us construct a starting policy $\pi_0$:

$\pi_0(s_1) = a_1$, $\pi_0(s_2) = a_2$, $\pi_0(s_3) = a_2$, $\pi_0(s_4) = a_2$. The other states are terminal and there are no actions to take, and therefor no policy.

Consider the following Policy Improvement operator $\mathcal{I}_E : \Pi \times \mathcal{Q} \to \Pi$, which this example will demonstrate is *not* a greedification operator. $\mathcal{I}_E$ takes a policy $\pi$, and mutates it with a random process to $\pi'$. $\mathcal{I}_E$ proceeds to conduct exact evaluation of $\pi'$, to find $V^{\pi'}$. If $V^{\pi'}(s) \geq V^\pi(s)$ on all states, and $V^{\pi'}(s) > V^\pi(s)$ in at least one state, $\mathcal{I}_E$ outputs $\pi'$. Otherwise, the process repeats. This process guarentees policy improvement. However, this process may directly produce the optimal policy in this MDP, which in states $s_1, s_3$ is $\pi^*(s_1) = a_3$ and $\pi^*(s_4) = a_1$.

Note however, that the optimal policy is *not* a greedier policy with respect to the value of $\pi_0$. For $\pi_0$, we have: $Q^{\pi_0}(s_1, a_1) = -1, Q^{\pi_0}(s_1, a_2) = 0, Q^{\pi_0}(s_1, a_3) = -2$. A greedier policy with respect to these values cannot deterministically choose action $a_3$, which is the action chosen by the optimal policy in this state.

Therefor, this example demonstrates that it is possible for a policy to be improved (higher value in at least one state, and greater or equal on all states), without being greedier with respect to some original policy's value. In turn, this demonstrates that there exist Policy Improvement operators that are *not* Greedification operators. $\square$

## A.3 Necessary greedification operators may not be sufficient

**Lemma 2.** *Greedification operators (Definition 2) which have the necessary greedification property (Definition 3) may not be sufficient for Policy Iteration algorithms to converge to the optimal policy.*

*Proof sketch:* First, we demonstrate the problem: certain operators with the necessary property, such as the deterministic greedification operators, may not converge to a greedy policy with respect to non-stationary $Q^{\pi_n}$. Second, we will show that in Policy Iteration it is possible to experience non stationary evaluations $Q^{\pi_n}$ that will prevent a necessary-greedification algorithm from converging to a greedy policy. We will show this by constructing an operator that performs deterministic

greedification in some states, and greedification that converges only in the limit in other states (both necessary-greedification operators), and show that the problem can persist in practical MDPs. This operator is a necessary-greedification operator in all states. That is, with respect a stationary $q$, it will converge to greedy policies. However, since in Policy Iteration algorithm the evaluation $q$ is not necessarily stationary, it is possible that a Policy Iteration algorithm based in this operator will not converge to the optimal policy.

*Proof.* Let $\mathcal{A} = \{a_1, a_2, a_3\}$ and a sequence $q_n(a_1) = (-1)^n/2^n + q(a_1)$, $q_n(a_2) = (-1)^{n+1}/2^n + q(a_2)$, and $q_n(a_3) = q(a_3)$, with a limiting value $q = [1, 1, 2]$. We omit the dependency of $q$ on a state as it is unnecessary in this example. In this case, the optimal policy with respect to any $q_n$ is $\pi = a_3$.

Take the least-greedifynig deterministic greedification operator $\mathcal{I}_{min\_det}(q, \pi) = \min_{q(a) > q(\pi), a \neq \pi} q(a)$. This operator produces the worst action, with respect to $q$, that is better than the current action selected by the policy, and as such, is a greedification operator by definition, with respect to deterministic policies. Since there are finitely many deterministic policies on a finite action space $|\mathcal{A}| < \infty$, this operator will converge to $\lim_{n \to \infty} \pi_n = \arg\max_a q(a)$ with respect to a stationary q.

Take $\pi_0 = a_2$. Using the operator $\mathcal{I}_{min\_det}$ we have $\pi_n = \mathcal{I}_{min\_det}(q_n, \pi_{n-1})$. When $n$ is odd, $\pi_n = a_1$, and when $n$ is even, $\pi_n = a_2$, without ever converging to the optimal policy $\pi = a_3$.

Next we will construct an example MDP and improvement operators in which this situation can happen in practice. Consider a finite-state, finite horizon MDP with states $s_1, \ldots, s_n$. We are interested in the behavior at state $s_0$ specifically, which similarly has actions $a_1, a_2, a_3$, with rewards $R(s_1, a_3) = 3, R(s_1, a_1) = R(s_1, a_2) = 0$. The transition $f(s_1, a_3) = s_0$ is terminal and $f(s_1, a_1) = s_2$, $f(s_1, a_2) = s_3$.

Consider the following improvement operator: On state $s_1$, this operator is $\mathcal{I}_{min_det}$. However, on all other states, this is a necessary greedification operator, which converges only in the limit, and in a non-constant rate. It is possible to construct the rest of the MDP and starting policies $\pi_0$ such that the sequence alternates $1 > Q^{\pi_n}(s_1, a_1) > Q^{\pi_n}(s_1, a_2)$ when $n$ is odd, and $1 > Q^{\pi_n}(s_1, a_2) > Q^{\pi_n}(s_1, a_1)$ when $n$ is even, while both are smaller than $Q^{\pi_n}(s_1, a_3) = Q^*(s_1, a_3) = 3$. This is possible because the policies $\pi_n(s_2), \pi_n(s_3)$ can be soft, and it is possible to construct an MDP which produces arbitrary values bounded between $0, 1$ by setting $R(s_2, a_1) = 1, R(s_2, a_2) = 0, R(s_2, a_3) = 0$ and $R(s_3, a_1) = 1, R(s_3, a_2) = 0, R(s_3, a_3) = 0$. In such MDP, $\lim_{n \to \infty} \pi_n(s_1)$ will never converge to $a_3$, the optimal policy in this state. $\qquad\square$

### A.4 Deterministic greedification operators are lower-bounded greedification operators

**Lemma 3.** *Deterministic greedification operators $\mathcal{I}_{det}$, i.e. greedification operators (Definition 2) that produce deterministic policies are lower-bounded greedification operators 4 in MDPs with finite action spaces.*

*Proof.* Take $\epsilon = \min_{s \in \mathcal{S}, a, a' \in \mathcal{A}, q(s,a) \neq q(s,a')} |q(s, a) - q(s, a')|$, that is, the minimum difference across two actions that do not have the same value (i.e. the minimum greater than zero difference). If there is no greater than zero difference, then all actions are optimal and every policy is already optimal. Otherwise, the greedification imposed by choosing at least one better action in at least one state has to be greater than the minimum difference between two actions. $\qquad\square$

### A.5 The operator $\mathcal{I}_{gmz}$ is a Limit-Sufficient Greedification operator

The operator proposed by Danihelka et al. (2022) is defined as follows:

$$\mathcal{I}_{gmz}(\pi, q)(a|s) = \text{softmax}(\sigma(q(s, a)) + \log \pi(a|s)) = \frac{\exp(\log \pi(a|s) + \sigma(q(s,a)))}{\sum_{a' \in \mathcal{A}} \exp(\log \pi(a'|s) + \sigma(q(s,a')))} \quad (9)$$

**Lemma 4** ($\mathcal{I}_{gmz}$ with a stationary $\sigma$ is a Limit-Sufficient Greedification Operator). *For any starting policy $\pi_0 \in \Pi$ such that $\log \pi_0(a|s)$ is defined and sequences $q_1, \ldots, q_n$ such that $\lim_{n \to \infty} q_n = q \in \mathcal{Q}$, iterative applications $\pi_{n+1} = \mathcal{I}_{gmz}(\pi_n, q_n)$ converge to a greedy policy with respect to the limiting value q.*
*That is,*

$$\lim_{n \to \infty} \sum_{a \in \mathcal{A}} \pi_n(a|s) q_n(s, a) = \max_b q(s, b), \quad \forall s \in \mathcal{S}.$$

*Proof sketch:* We will prove that $n$ repeated applications of the $\mathcal{I}_{gmz}$ operator converge to a softmax policy of the form

$$\pi_n(a|s) \propto \exp(\log \pi_0(a|s) + n\sigma(q_n(s,a))),$$

which itself converges to an $\arg\max$ policy as $\lim_{n\to\infty}$. For simplicity, we will first prove for a stationary $q$, and then repeat the same steps for a non-stationary $q_n, \lim_{n\to\infty} = q$ for some limiting value $q$.

*Proof.* We will show that the Gumbel MuZero operator $\mathcal{I}_{gmz}$ with $\sigma$ a monotonically increasing transformation, is a Limit-Sufficient Greedification operator.

Danihelka et al. (2022) have shown that this operator is a Greedification operator (Section 4 and Appendix C of (Danihelka et al., 2022)). It remains for us to demonstrate that the sequence $\pi_n$ converges for $\mathcal{I}_{gmz}$, such that

$$\lim_{n\to\infty} \sum_{a\in\mathcal{A}} \pi_n(a|s)q(s,a) = \max_b q(s,a),$$

for any $\pi_0$ and $\forall s \in \mathcal{S}$.

**Step 1: Convergence with stationary** $q$    For a stationary $q$, at any iteration $n$, the policy $\pi_n$ can be formulated as:

$$\pi_n(a) = \frac{1}{z_n} \exp(\sigma(q(s,a)) + \log \pi_{n-1}(a|s)), \quad z_n = \sum_{a'\in\mathcal{A}} \exp(\sigma(q(s,a')) + \log \pi_{n-1}(a'|s)) \tag{10}$$

Where $z_n$ is the normalizer of the $\mathrm{softmax}$ operator. We can expand $\pi_n$ backwards as follows:

$$\pi_n(a) = \frac{1}{z_n} \exp(\sigma(q(s,a)) + \log \pi_{n-1}(a|s)) \tag{11}$$

$$= \frac{1}{z_n} \exp\left(\sigma(q(s,a)) + \log \frac{\sigma(q(s,a)) + \pi_{n-1}(a|s)}{z_{n-1}}\right) \tag{12}$$

$$= \frac{1}{z_n} \exp\left(\sigma(q(s,a)) + \sigma(q(s,a)) + \log \pi_{n-2}(a|s) - \log z_{n-1}\right) \tag{13}$$

$$= \frac{1}{z_n z_{n-1}} \exp\left(2\sigma(q(s,a)) + \log \pi_{n-2}(a|s)\right) \tag{14}$$

$$= \dots \tag{15}$$

$$= (\Pi_{i=1}^n \frac{1}{z_i}) \exp\left(n\sigma(q(s,a)) + \log \pi_0(a|s)\right) \tag{16}$$

As $\pi_n$ is a softmax policy, i.e. $\sum_{a\in\mathcal{A}} \pi_n(a|s) = 1$, the product $\Pi_{i=1}^n \frac{1}{z_i}$ must act as a normalizer:

$$\Pi_{i=1}^n \frac{1}{z_i} = \sum_{a\in\mathcal{A}} \exp\left(n\sigma(q(s,a)) + \log \pi_0(a|s)\right) \tag{17}$$

We can now directly take the limit $\lim_{n\to\infty} \pi_n$:

$$\lim_{n\to\infty} \pi_n(a) = \lim_{n\to\infty} (\Pi_{i=1}^n \frac{1}{z_i}) \exp\left(n\sigma(q(s,a)) + \log \pi_0(a|s)\right) \tag{18}$$

It is well established that as the temperature $1/n$ goes to zero, the $\mathrm{softmax}$ converges to an $\arg\max$ (Collier et al., 2020). With non-stationary $q_n$ we get a slightly more involved sequence, and the formulated proof that the $\mathrm{softmax}$ converges to an $\arg\max$ will serve us to demonstrate convergence with $q_n$. We include the proof that the $\mathrm{softmax}$ converges to an $\arg\max$ below in step 1.5.

**Step 1.5: Convergence of the** $\mathrm{softmax}$ **to an** $\arg\max$    Define $\sigma_{max} = \max_a \sigma(q(s,a))$. Let us now multiply by $\frac{\exp(-n\sigma_{max})}{\exp(-n\sigma_{max})}$. We have:

$$\pi_n(a|s) = (\Pi_{i=1}^n \frac{1}{z_i}) \exp\left(n\sigma(q(s,a) + \log \pi_0(a|s))\right) \frac{\exp(-n\sigma_{max})}{\exp(-n\sigma_{max})} \tag{19}$$

$$= \frac{\pi_0(a|s)}{\exp(-n\sigma_{max})} (\Pi_{i=1}^n \frac{1}{z_i}) \exp\left(n(\sigma(q(s,a)) - \sigma_{max})\right) \tag{20}$$

We now note that $\sigma(q(s,a)) - \sigma_{max} < 0$ if $\sigma(q(s,a)) \neq \max_a \sigma(q(s,a)) = \sigma_{max}$ and otherwise $\sigma(q(s,a)) - \sigma_{max} = 0$ if $\sigma(q(s,a)) = \max_a \sigma(q(s,a)) = \sigma_{max}$. In that case, $\exp\big(n(\sigma(q(s,a)) - \sigma_{max})\big) = \exp(0) = 1$. We substitute that into the limit:

$$\lim_{n \to \infty} \pi_n(a|s) = \begin{cases} \lim_{n \to \infty} \frac{\pi_0(a|s)}{\exp(-n\sigma_{max})}(\Pi_{i=1}^n \frac{1}{z_i})\exp\Big(n\big(\sigma(q(s,a)) - \sigma_{max}\big)\Big) = 0, & \text{if } \sigma(q(s,a)) \neq \sigma_{max} \\ \lim_{n \to \infty} \frac{\pi_0(a|s)}{\exp(-n\sigma_{max})}(\Pi_{i=1}^n \frac{1}{z_i})1, & \text{if } \sigma(q(s,a)) = \sigma_{max} \end{cases}$$

(21)

Note that:

1. The numerator where $\sigma(q(s,a)) = \sigma_{max}$ converges to $e^0 = 1$

2. The numerator where $\sigma(q(s,a)) \neq \sigma_{max}$ converges to $\lim_{n \to \infty} e^{-\delta n} = 0, \delta > 0$.

3. The denominator always normalizes the policy such that $\sum_{a \in \mathcal{A}} \pi_n(a|s) = 1, \forall s \in \mathcal{S}$, due to the definition of the softmax.

As a result, we have:

$$\lim_{n \to \infty} \pi_n(a|s) = \begin{cases} \frac{0}{z}, & \text{if } \sigma(q(s,a)) \neq \sigma_{max} \\ \frac{1}{z}, & \text{if } \sigma(q(s,a)) = \sigma_{max} \end{cases}$$

(22)

For some normalization constant $z$. I.e. the policy $\lim_{n \to \infty} \pi_n$ is an $\arg\max$ policy with respect to $q$, that is, the policy has probability mass only over actions that maximize $\sigma(q)$.

**Step 2: Convergence with non-stationary $q_n$** We will now extend the proof to a non-stationary $q_n$ that is assumed to have a limiting value, $\lim_{n \to \infty} q_n = q$, in line with definition of Sufficient Greedification.

First, we have:

$$\pi_n(a) = \frac{1}{z_n} \exp(\sigma(q_n(s,a)) + \log \pi_{n-1}(a|s)) \tag{23}$$

$$= \frac{1}{z_n} \exp\Big(\sigma(q_n(s,a)) + \log \frac{\sigma(q_{n-1}(s,a)) + \pi_{n-1}(a|s)}{z_{n-1}}\Big) \tag{24}$$

$$= \frac{1}{z_n z_{n-1}} \exp\Big(\sigma(q_n(s,a)) + \sigma(q_{n-1}(s,a)) + \log \pi_{n-2}(a|s)\Big) \tag{25}$$

$$= (\Pi_{i=1}^n \frac{1}{z_i}) \exp\Big(\big(\sum_{i=1}^n \sigma(q_i(s,a))\big) + \log \pi_0(a|s)\Big) \tag{26}$$

Based on the same expansion of the sequence as above. Multiplying by $\frac{-n\sigma_{max}}{-n\sigma_{max}}$ and formulating the limit in a similar manner to above, we then have:

$$\lim_{n \to \infty} \pi_n(a|s) = \lim_{n \to \infty} \frac{\pi_0(a|s)}{-n\sigma_{max}} \exp(\Pi_{i=1}^n \frac{1}{z_i})\big(\sum_{i=1}^n (\sigma(q_i(s,a)) - \sigma_{max})\big) \tag{27}$$

Let us look at the term $\sum_{i=1}^n (\sigma(q_i(s,a)) - \sigma_{max})$. First, where $\sigma(q(s,a)) \neq \sigma_{max}$, we have

$$\lim_{n \to \infty} \sum_{i=1}^n (\sigma(q_i(s,a)) - \sigma_{max}) = \lim_{n \to \infty} \sum_{i=1}^n (\sigma(q_i(s,a)) - \sigma(q(s,a)) - (\sigma_{max} - \sigma(q(s,a)))) \tag{28}$$

$$= \lim_{n \to \infty} -n(\sigma_{max} - \sigma(q(s,a))) + \sum_{i=1}^n (\sigma(q_i(s,a)) - \sigma(q(s,a))) \tag{29}$$

As the term $\sigma(q_n(s,a)) - \sigma(q(s,a))$ goes to zero due to the definition of $q_n$, the term $\sum_{i=1}^n (\sigma(q_i(s,a)) - \sigma(q(s,a)))$ goes to a constant, and the term $-n(\sigma_{max} - \sigma(q(s,a)))$ goes to $-\infty$ due to the definition of $\sigma_{max}$. Therefor, the limit:

$$\lim_{n \to \infty} \sum_{i=1}^n (\sigma(q_i(s,a)) - \sigma_{max}) = -\infty \quad \Rightarrow \quad \lim_{n \to \infty} \exp\big(\sum_{i=1}^n (\sigma(q_i(s,a)) - \sigma_{max})\big) = 0 \tag{30}$$

Let us look at the second case, where $\sigma(q(s,a)) = \sigma_{max}$, the sequence converges:

$$\lim_{n \to \infty} \sum_{i=1}^{n} (\sigma(q_i(s,a)) - \sigma_{max}) = \alpha(s,a) \tag{31}$$

For some constant $\alpha(s,a)$, as $\lim_{n \to \infty} \sigma(q_n(s,a)) = \sigma_{max}$. Thus, we have again:

$$\lim_{n \to \infty} \pi_n(a|s) = \begin{cases} \frac{0}{z}, & \text{if } \sigma(q(s,a)) \neq \sigma_{max} \\ \frac{\alpha(s,a)}{z}, & \text{if } \sigma(q(s,a)) = \sigma_{max} \end{cases} \tag{32}$$

Demonstrating that $\pi_n$ converges to an $\arg\max$ policy with respect to $\sigma(q)$. Since $\sigma$ is monotonically increasing, $q(s,a)$ and $\sigma(q(s,a))$ are maximized for the same action $a$, thus $\pi_n$ is also an $\arg\max$ policy with respect to $q$. Therefor, $\mathcal{I}_{gmz}$ is a Limit-Sufficient Greedification operator. $\qquad\square$

## A.6 $\mathcal{I}_{gmz}$ can be formulated as an insufficient-greedification operator

**Lemma 5.** *The $\mathcal{I}_{gmz}$ greedification operator with a non-stationary $\sigma$ transformation can be formulated as an insufficient greedification operator.*

*Proof sketch:* We construct a variation of the $\mathcal{I}_{gmz}$ operator with an increasing transformation $\sigma_n$, which is different at each iteration. Because the transformation is not constant, it converges to some softmax policy rather than an $\arg\max$ policy.

*Proof.* The function $\sigma$ used by $\mathcal{I}_{gmz}$ is only required to be an increasing transformation (see Danihelka et al. (2022), Section 3.3). That is if $q(s,a) > q(s,a')$ then we must have that $\sigma(q(s,a)) > \sigma(q(s,a'))$. In practice, the function proposed by Danihelka et al. (2022) is of the form $\sigma(q(s,a)) = \beta(N)q(s,a)$, where $\beta$ is a function of the planning budget $N$ of the MCTS algorithm. A practitioner might be interested in running the algorithm with a decreasing planning budget over iterations (perhaps the value estimates become increasingly more accurate, and therefor there is less reason to dedicate much compute into planning with MCTS). In that case, we can formulate $\sigma_n(q_n(s,a)) = \frac{\alpha}{\beta^n} q_n(s,a)$. This transformation is always increasing in $q(s,a)$, adhering to the requirements from $\sigma$. Nonetheless, the sequence $\pi_n$ will not converge to an argmax policy for this choice of $\sigma$:

$$\lim_{n \to \infty} \pi_n = \lim_{n \to \infty} (\Pi_{i=1}^{n} \frac{1}{z_i}) \exp\Big(\sum_{i \leq n} \big[\sigma_n(q_n(s,a))\big] + \log \pi_0(a|s)\Big) \tag{33}$$

$$= \lim_{n \to \infty} (\Pi_{i=1}^{n} \frac{1}{z_i}) \exp\Big(\frac{\alpha}{\beta^n} \sum_{i \leq n} \big[q_n(s,a)\big] + \log \pi_0(a|s)\Big) \tag{34}$$

Which will converge to some softmax policy as the following limit converges to a constant: $\lim_{n \to \infty} \frac{\alpha}{\beta^n} \sum_{i \leq n} \big[q_n(s,a)\big] = c(s,a)$, and thus the policy remains a softmax policy $\pi_n(a|s) = \text{softmax}(c(s,a) + \log \pi_0(a|s))$. $\qquad\square$

## A.7 Lower Bounded Greedification operators $\not\subset$ Limit-Sufficient Greedification operators

**Lemma 6.** *The set of all lower-bounded greedification operators (Definition 4) is not a subset of the set of all limit-sufficient greedification operators (Definition 5). That is, there exists a lower-bounded greedification operator which is not a limit-sufficient greedification operator.*

*Proof sketch:* Convergence with respect to arbitrary sequences $\lim_{n \to \infty} q_n = q$ is a strong property, and it is possible to come up with sequences for which specific lower-bounded greedification operator do not result in convergence. By constructing such a sequence and choosing such an operator, we will show that there are lower-bounded greedification operators which are not Limit-Sufficient Greedification operators, demonstrating that lower-bounded greedification operators $\not\subset$ limit-sufficient greedification operators.

*Proof.* Let $\mathcal{A} = \{a_1, a_2, a_3\}$ and a sequence $q_n(a_1) = (-1)^n/2^n + q(a_1)$, $q_n(a_2) = (-1)^{n+1}/2^n + q(a_2)$, and $q_n(a_3) = q(a_3)$, with a limiting value $q = [1,1,2]$.

Let $\pi_0 = a_1$. The minimal deterministic Greedification operator $\mathcal{I}_{det}(\pi, q)(s) = \arg\min_a q(s,a) > \sum_{a' \in \mathcal{A}} \pi(a'|s)q(s,a')$, that is, the deterministic Greedification operator which chooses the least-greedifying action at each step will not converge to the optimal policy on this sequence. At each

iteration, $\mathcal{I}_{det}(q, \pi_n) = a_{1,2}$ (as in, $a_1$ or $a_2$), because $q_n$ alternates $q_n(a_1) > q_n(a_2)$ for even $n$, and $q_n(a_1) < q_n(a_2)$ for odd $n$. Since this operator is a lower-bounded greedification operator (see Appendix A.4), this demonstrates that lower-bounded greedification operators $\not\subset$ limit-sufficient greedification operators. $\qquad\square$

## A.8 The greedy operator is both a limit-sufficient as well as a lower-bounded greedification operator

**Lemma 7.** *The greedy operator $\mathcal{I}_{\arg\max}$ is both a lower-bounded greedification operator (Definition 4) as well as a limit-sufficient greedification operator (Definition 5).*

*Proof.* The greedy operator is a greedification operator by definition. We will show that it can have both the lower-bounded greedification property as well as the limit sufficient greedification property.

Step 1): We will show that the greedy operator is a lower-bounded greedification operator (Definition 4).

The greedy operator produces the maximum greedification in any state by definition. Therefor:

$$\sum_{a \in A} \mathcal{I}_{argmax}(\pi, q)(a|s)q(s, a) \geq \mathcal{I}_{det}(\pi, q)(a|s)q(s, a),$$

where $I_{det}$ is the deterministic greedification operator, $\forall s \in \mathcal{S}, a \in \mathcal{A}$. Since the deterministic greedification operator is itself bounded by an $\epsilon$ (see Appendix A.4), we have $|\sum_{a \in A} \mathcal{I}_{argmax}(\pi, q)(a|s)q(s, a) - \sum_{a \in A} \pi(a|s)q(s, a)| > \epsilon$.

Step 2): We will show that the greedy operator is a limit-sufficient greedification operator (Definition 5).

We will prove that the sequence $(\pi_n, q_n)$ defined for $\mathcal{I}_{\arg\max}$ as above converges, such that $\lim_{n \to \infty} |\sum_{a \in \mathcal{A}} \pi_n(a|s)q_n(s, a) - \max_b q(s, b)| = 0$, for any $\pi_0$. That is, the policy converges to an $\arg\max$ policy with respect to the limiting value $q$.

For any $q_n$ in the sequence, we have by definition of the operator $\sum_{a \in \mathcal{A}} \mathcal{I}_{\arg\max}(q_n, \pi_{n-1})(a|s)q_n(s, a) = \max_a q_n k(s, a)$. We can substitute that into the limit:

$$\lim_{n \to \infty} |\sum_{a \in \mathcal{A}} \pi_n(a|s)q_n(s, a) - \max_b q(s, b)| \tag{35}$$

$$= \lim_{n \to \infty} |\max_a q_n(s, a) - \max_b q(s, b)| \tag{36}$$

$$\leq \lim_{n \to \infty} \max_a |q_n(s, a) - q(s, a)| \tag{37}$$

$$= \max_a \lim_{n \to \infty} |q_n(s, a) - q(s, a)| \tag{38}$$

$$= \max_a |\lim_{n \to \infty} q_n(s, a) - q(s, a)| = 0 \tag{39}$$

The first step holds by substitutions. The inequality is a well known property used to prove that the greedy operator is a contraction, see (Blackwell, 1965). In Equation 38 the limit and max operators can be exchanged because the action space is finite, and finally the limit and absolute value can be exchanged because the absolute value is a continuous function.

$\qquad\square$

## A.9 Proof for Theorem 3 for $k = 1$ and $\mathcal{I}_2$ the identity operator

We will prove Theorem 3, first for $k = 1$ for improved readability, and in the following Appendix A.10 we will extend the proof for $k \geq 1$. In Appendix A.11 we will further extend the proof for value-improvement.

### A.9.1 Notation

We use $\mathcal{R}$ to denote the mean-reward vector $\mathcal{R} \in \mathbb{R}^{|\mathcal{S}||\mathcal{A}|}$, where $\mathcal{R}_{s,a} = \mathbb{E}[R|s, a]$. We use $\mathcal{P}^{\pi} \in \mathbb{R}^{|\mathcal{S}||\mathcal{A}| \times |\mathcal{S}||\mathcal{A}|}$ to denote the matrix of transition probabilities multiplied by a policy, indexed as follows: $\mathcal{P}^{\pi}_{s,a,s',a'} = P(s'|s, a)\pi(a'|s')$. We denote the state-action value $q$ and the policy $\pi$ as vectors in the state-action space s.t. $q, \pi \in \mathbb{R}^{|\mathcal{S}||\mathcal{A}|}$. The set $\Pi \subset R^{|\mathcal{S}||\mathcal{A}|}$ contains all admissible policies that define a probability distribution over the action space for every state. For convenience,

we denote $q(s, a)$ as a specific entry in the vector indexed by $s, a$ and $q(s), \pi(s)$ as the appropriate $|\mathcal{A}|$ dimensional vectors for index $s$. In this notation, we can write expectations as the dot product $q(s) \cdot \pi(s) = \mathbb{E}_{a \sim \pi(s)}[q(s, a)] = v(s)$. With slight abuse of notation, we use $q \cdot \pi = v$, $v \in \mathbb{R}^{|\mathcal{S}|}$ to denote the vector with entries $v(s)$. We use $\max_a q \in \mathbb{R}^{|\mathcal{S}|}$ to denote the vector with entries $\max_a q(s) = \max_a q(s, a)$.

We let $s_t$ denote a state $(\cdot, t) \in \mathcal{S}$, that is, a state in the environment arrived at after $t$ transitions. The states $s_H$ are terminal states, and the indexing begins from $s_0$. We let $q^m, \pi^m$ denote the vectors at iteration $m$ of Algorithm 1. We let $q_t^m, \pi_t^m$ denote the sub-vectors of all entries in $q^m, \pi^m$ associated with states $s_t$. In this notation $q_{H-1}^1$ is the $q$ vector for all terminal transitions $(s_{H-1}, \cdot)$ after the one iteration of the algorithm.

**Proof sketch** Our proof follows induction from terminal states. For all terminal states $s_H$, the value $V^\pi(s_H) = 0$ for all policies $\pi$. Similarly, $q(s_{H-1}, a) = Q^\pi(s_{H-1}, a) = r(s, a)$ for all policies $\pi$. That is, the Q values converge after one update, and from then on remain stationary. Given that the $q(s_{H-1}, a)$ remains stationary for all states $s_{H-1}$, limit sufficient greedification guarantees that policy $\pi(s_{H-1})$ at state $s_{H-1}$ converges to an $\arg\max$ policy, which guarantees that the state-value estimates $v(s_{H-1}) := \sum_{a \in \mathcal{A}} \pi(a|s_{H-1})q(s_{H-1}, a)$ converge. Finally, as the state-value estimates converge, this process repeats backwards from states $s_{H-1}$ all the way to states $s_0$, at which point the value $q$ and policy $\pi$ converge to the value of the optimal policy and an optimal policy respectively, for all states in the MDP.

### A.9.2 Complete proof

*Proof.* Convergence for Generalized Policy Iteration with $k = 1$

We will prove by backwards induction from the terminal states that the sequence $\lim_{m \to \infty}(\pi^m, q^m)$ induced by Algorithm 1 converges for any $q^0, \pi^0$, sufficient greedification operator $\mathcal{I}$ and $k \geq 1$. That is, for every $\epsilon > 0$ there exists a $M_\epsilon$ such that $\|q^m - q^*\| \leq \epsilon$ and $\|\pi^m \cdot q^m - \max_a q^*\| < \epsilon$ for all $m \geq M_\epsilon$, $q^0 \in \mathbb{R}^{|\mathcal{S}||\mathcal{A}|}$ and $\pi^0 \in \Pi$.

**Induction Hypothesis:** For every $\epsilon > 0$ there exist $M_{t+1}^\epsilon$ such that for all $m \geq M_{t+1}^\epsilon$ we have $\|q_{t+1}^m - q_{t+1}^*\| \leq \epsilon$, and $\|\pi_{t+1}^m \cdot q_{t+1}^m - \max_a q_{t+1}^*\| \leq \epsilon$.

**Base Case** $t = H - 1$: Let $\epsilon > 0$. Since states $s_H$ are terminal, and have therefore value 0, we have $q_{H-1}^m = \mathcal{R}_{H-1} = q_{H-1}^*$ and therefore $\|q_{H-1}^m - q_{H-1}^*\| \leq \epsilon$ trivially holds for all $m \geq 1$.

By the Sufficiency condition of the sufficient greedification operator which induces convergence of $\pi^m$ to an $\arg\max$ policy with respect to $q$ there exists $M_{H-1}^\epsilon$ such that:

$$\|\pi_{H-1}^m \cdot q_{H-1}^m - \max_a q_{H-1}^*\| = \|\pi_{H-1}^m \cdot q_{H-1}^* - \max_a q_{H-1}^*\| \leq \epsilon$$

for all $m \geq M_{H-1}^\epsilon$. Thus the Induction Hypothesis holds at the base case.

**Case** $t < H - 1$: We will show that if the Induction Hypothesis holds for all states $t + 1$, it also holds for states $t$.

*Step 1:* Let $\epsilon > 0$. Assume the Induction Hypothesis holds for states $t + 1$. Then there exists $M_{t+1}^\epsilon$ such that $\|q_{t+1}^m - q_{t+1}^*\| \leq \epsilon$ and $\|\pi_{t+1}^m \cdot q_{t+1}^m - \max_a q_{t+1}^*\| \leq \epsilon$ for all $m \geq M_{t+1}^\epsilon$.

Let us define the transition matrix $\mathcal{P} \in \mathbb{R}^{|\mathcal{S}||\mathcal{A}| \times |\mathcal{S}|}$ with $\mathcal{P}_{s,a,s'} = P(s'|s, a)$.

First, for all $m \geq M_{t+1}^\epsilon$ we have:

$$\|q_t^{m+1} - q_t^*\| = \|\mathcal{R} + \gamma \mathcal{P}(\pi_{t+1}^{m+1} \cdot q_{t+1}^m) - \mathcal{R} - \gamma \mathcal{P} \max_a q_{t+1}^*\| \tag{40}$$

$$= \gamma \|\mathcal{P}(\pi_{t+1}^{m+1} \cdot q_{t+1}^m) - \mathcal{P} \max_a q_{t+1}^*\| \tag{41}$$

$$\leq \|\mathcal{P}\| \|\pi_{t+1}^{m+1} \cdot q_{t+1}^m - \max_a q_{t+1}^*\| \tag{42}$$

$$\leq \|\pi_{t+1}^{m+1} \cdot q_{t+1}^m - \max_a q_{t+1}^*\| \tag{43}$$

$$\leq \epsilon \tag{44}$$

(40) is by substitution based on step 4 in Algorithm 1 for $k = 1$. (42) is by the definition of the operator norm $\|\mathcal{P}\|$. (43) is by the fact that the operator norm in sup-norm of all transition matrices is 1 (Bertsekas, 2007). (44) is slightly more involved, and follows from the Induction Hypothesis and the limit-sufficient greedification.

Let us show that (44), i.e. $\|\pi_{t+1}^{m+1} \cdot q_{t+1}^m - \max_a q_{t+1}^*\| \le \epsilon$ holds. Under the infinity norm holds point-wise for each state $s \in \mathcal{S}$:

$$-\epsilon \le [\pi_{t+1}^m \cdot q_{t+1}^m](s) - \max_a q_{t+1}^*(s, a) \tag{45}$$

$$\le [\boldsymbol{\pi_{t+1}^{m+1} \cdot q_{t+1}^m}](\boldsymbol{s}) - \max_{\boldsymbol{a}} \boldsymbol{q_{t+1}^*(s, a)} \tag{46}$$

$$\le \max_{a'} q_{t+1}^m(s, a') - \max_a q_{t+1}^*(s, a) \tag{47}$$

$$\le \max_{a'} \left( q_{t+1}^m(s, a') - q_{t+1}^*(s, a') \right) \tag{48}$$

$$\le \epsilon. \tag{49}$$

(45) is the induction hypothesis $\|\pi_{t+1}^m \cdot q_{t+1}^m - \max_a q_{t+1}^*\| \le \epsilon$, which holds under the infinity norm point wise, (46) uses the sufficient greedification operatorproperty $[\pi_{t+1}^m \cdot q_{t+1}^m](s) \le [\pi_{t+1}^{m+1} \cdot q_{t+1}^m](s)$, (47) the inequality $[\pi_{t+1}^{m+1} \cdot q_{t+1}^m](s) \le \max_{a'} q_{t+1}^m(s, a')$, (48) the inequality $-\max_a q_{t+1}^*(s, a) \le -q_{t+1}^*(s, a'), \forall a' \in \mathcal{A}$, and (49) the induction hypothesis $\|q_{t+1}^* - q_{t+1}^m\| \le \epsilon$.

*Step 2:* Pick $M_t^\epsilon \ge M_{t+1}^\epsilon$ such that for all $m \ge M_t^\epsilon$ we have $\|\pi_t^m \cdot q_t^m - \max_a q_t^*\| \le \epsilon$. Such an $M_t^\epsilon$ must exist because of the following: In Step 1, we proved that the first part of the inductive step holds. That is, that $q_t^m$ has $\epsilon$-converged to the value of the optimal policy. Such $q_t^m$ satisfies the conditions of the $q$ sequence of the limit-sufficient greedification operator. For that reason, the policy $\pi_t^m$ must converge (that is $\|\pi_t^m \cdot q_t^m - \max_a q_t^*\| \le \epsilon$), and $M_t^\epsilon$ must exist.

Thus, the Induction Hypothesis holds for all states $t$ if it holds for states $t + 1$.

Finally, let $\epsilon > 0$. By backwards induction, for each $t = 0, \ldots, H - 1$ there exists $M_\epsilon^t$ such that for all $m \ge M_\epsilon^t$ we have $\|q_t^m - q_t^*\| \le \epsilon$, and $\|\pi_t^m \cdot q_t^m - \max_a q_t^*\| \le \epsilon$. Therefore, we can pick $N_\epsilon = \max_{t=0,\ldots,H-1} M_\epsilon^t$ such that $\|q_t^m - q_t^*\| \le \epsilon$, and $\|\pi_t^m \cdot q_t^m - \max_a q_t^*\| \le \epsilon$ for all $m \ge N_\epsilon$ and $t = 0, \ldots, H - 1$, proving that Algorithm 1 converges to an optimal policy and optimal q-values for any $\pi^0 \in \Pi, q^0 \in \mathbb{R}^{|\mathcal{S}||\mathcal{A}|}, k = 1$ and sufficient greedification operator$\mathcal{I}$. $\qquad\square$

We proceed to extend the proof for $k \ge 1$ below.

## A.10   Extension of the Proof for Theorem 3 to $k \ge 1$ and $\mathcal{I}_2$ the identity operator

In this section we will extend the proof of Theorem 3 from Appendix A.9 to $k \ge 1$. Much of the proof need not be modified. In order to extend the proof to $k \ge 1$, we only need to show the following: For all $k \ge 1$ and every $\epsilon > 0$ such that the Induction Hypothesis holds, there exists an $M_\epsilon^t$ such that $\|q_t^{m+1} - q_t^*\| \le \epsilon$.

*Proof.* We will first extend the notation: let $q_t^{m,i}$ denote the vector $q$ at states $t$ after $m$ algorithm iterations and $i \ge 1$ Bellman updates, such that $q_t^{m,i} = (\mathcal{T}^{\pi_1} q^{m,i-1})_t$, $q_t^{m,0} = q_t^m$ and finally $q_t^{m+1} = q_t^{m,k}$.

Second, we will extend the Induction Hypothesis:

**Extended Induction Hypothesis:** For every $\epsilon > 0$ there exist $M_{t+1}^\epsilon$ such that for all $m \ge M_{t+1}^\epsilon$ and $i \ge 0$ we have $\|q_{t+1}^{m,i} - q_{t+1}^*\| \le \epsilon$, and $\|\pi_{t+1}^m \cdot q_{t+1}^{m,i} - \max_a q_{t+1}^*\| \le \epsilon$.

The Base Case does not change, so we will proceed to *Step 1* in the Inductive Step. We need to show that there exists an $M_t^\epsilon$ such that $\|q_t^{m,i} - q_t^*\| \le \epsilon$ for all $i \ge 0$ and $m \ge M_t^\epsilon$.

Let $\epsilon > 0$ and $m \ge M_t^\epsilon \ge M_{t+1}^\epsilon$.

First, for any $i \ge 1$:

$$\|q_t^{m,i} - q_t^*\| = \|\mathcal{R} + \gamma \mathcal{P}(\pi_{t+1}^{m+1} \cdot q_{t+1}^{m,i-1}) - q_t^*\|$$

$$\le \|\mathcal{P}\| \|\pi_{t+1}^{m+1} \cdot q_{t+1}^{m,i-1} - \max_a q_{t+1}^*\|$$

$$\le \epsilon$$

The first equality is the application of the Bellman Operator in line 4 in Algorithm 1 the $i$th time. The rest follows from Proof A.9 and the extended Induction Hypothesis.

Second, we need to show that this holds for $i = 0$ as well:

$$\|q_t^{m,0} - q_t^*\| = \|q_t^{m-1,k} - q_t^*\| \le \|\pi_{t+1}^m \cdot q_{t+1}^{m-1,k-1} - \max_a q_{t+1}^*\| \le \epsilon$$

The first equality is by definition, and the the first and second inequalities are by the same argumentation as above.

The rest of the proof need not be modified. □

## A.11 Extension for $\mathcal{I}_2$ a general improvement operator

We extend the proof from the above section for all non-detriment operators (that is, non-strict greedification operators) $\mathcal{I}_2$ used for value improvement.

*Proof.* Similarly to the proof of Theorem 3 from Appendix A.9 (and A.10) we will prove by backwards induction from the terminal states $s_H$ that the sequence $\lim_{m\to\infty}(\pi^m, q^m)$ induced by Algorithm 2 converges for any $q^0, \pi^0$, sufficient greedification operator $\mathcal{I}_1$, greedification operator $\mathcal{I}_2$ and $k \geq 1$. That is, for every $\epsilon > 0$ there exists a $M_\epsilon$ such that $\|q^m - q^*\| \leq \epsilon$ and $\|\pi^m \cdot q^m - \max_a q^*\| < \epsilon$ for all $m \geq M_\epsilon$, $q^0 \in \mathbb{R}^{|\mathcal{S}||\mathcal{A}|}$ and $\pi^0 \in \Pi$. The proof follows directly from the proof in Appendix A.9. The base case is not modified - the $q$s converge immediately and the policy convergence is not influenced by the introduction of $\mathcal{I}_2$. The Induction Hypothesis need not be modified. In the inductive step, *Step 1* follows directly from the Induction Hypothesis, and *Step 2* need not be modified for the same reason the base case need not be modified. □

## A.12 Convergence of Algorithm 2 with lower bounded greedification operators

We extend the proof from Appendices A.9 and A.10 to bounded greedification operators.

**Proof sketch** The proof is almost identical to that of limit-sufficient greedification, with one major difference. Lower-bounded greedification allows for convergence to a greedy policy in *finite* iterations (see Lemma 8 below) with respect to any *stationary* $q$. For that reason, the values at states $s_{H-1}$ become exact in finite iterations (unlike limit-sufficient, where they converge only in the limit). As they become exact, they also become fully stationary, and as they become stationary, lower-bounded greedification guarantees that the values (and policy) at states $s_{H-2}$ become exact and stationary in finite iterations, and the process repeats by induction all the way back to states $s_0$.

Below we first prove Lemma 8 and then use Lemma 8 to complete the induction proof.

### A.12.1 Lower bounded greedification converges to an $\arg\max$ policy in finite steps

We will begin by proving that operators with the Bounded Greedification property:

$$\left| \sum_{a\in\mathcal{A}} \mathcal{I}(\pi, q)(a|s)q(s,a) - \sum_{a\in\mathcal{A}} \pi(a|s)q(s,a) \right| > \epsilon,$$

unless $\sum_{a\in\mathcal{A}} \mathcal{I}(\pi, q)(a|s)q(s,a) = \max_a q(s,a)$ are guaranteed to convergence to an $\arg\max$ policy with respect to any $q \in \mathcal{Q}$, in a finite number of steps.

**Lemma 8.** *Let $\mathcal{I}$ be a bounded greedification operator and let a sequence $\pi_{n+1} = \mathcal{I}(q, \pi_n)$. For any starting $\pi_0 \in \Pi, q \in \mathcal{Q}$, there exists an $M$ for which:*

$$\sum_{a\in\mathcal{A}} \pi_n(a|s)q(s,a) = \max_a q(s,a), \quad \forall n > M.$$

*That is, the policy $\pi_n$ converges to a greedy policy with respect to $q$ in a finite number of steps $n > M$.*

*Proof.* Let $\mathcal{I}$ be a Bounded Greedification operator. At each iteration, the sequence $\sum_{a\in\mathcal{A}} \pi_n(a|s)q(s,a)$ must increase, i.e. $\sum_{a\in\mathcal{A}} \pi_n(a|s)q(s,a) > \sum_{a\in\mathcal{A}} \pi_{n-1}(a|s)q(s,a), n > 0$, for at least one state $s \in \mathcal{S}$. The same sequence is monotonically non-decreasing, by definition of greedification, for all other states. Therefore, the sequence $\sum_{s\in\mathcal{S}} \sum_{a\in\mathcal{A}} \pi_n(a|s)q(s,a)$ is monotonically increasing (for each state $\sum_{a\in\mathcal{A}} \pi_n(a|s)q(s,a)$ is at least as large as in the past step, and in at least one state it is distinctly higher), unless $\sum_{a\in\mathcal{A}} \pi_n(a|s)q(s,a) = \max_a q(s,a)$.

Due to the Bounded Greedification property, the minimum increase is bounded by $\epsilon$, that is:

$$\min_{\pi_n} \left| \sum_{s\in\mathcal{S}} \sum_{a\in\mathcal{A}} \pi_n(a|s)q(s,a) - \sum_{s\in\mathcal{S}} \sum_{a\in\mathcal{A}} \pi_{n-1}(a|s)q(s,a) \right| > \epsilon, \quad n > 0.$$

The sequence $\sum_{s \in \mathcal{S}} \sum_{a \in \mathcal{A}} \pi_n(a|s) q(s,a)$ is bounded by $\sum_{s \in \mathcal{S}} \max_a q(s,a)$ from above, and by $\sum_{s \in \mathcal{S}} \min_a q(s,a)$ from below. The increases between any two iterations is bounded from below by $\epsilon$ by definition unless the policy is already greedy, as $I$ is a lower-bounded greedification operator.

Since the sequence is bounded from below and above and the increase is bounded a constant amount $\epsilon > 0$, it must converge in a finite $n < \infty$ to the maximum of the sequence $\sum_{a \in \mathcal{A}} \pi_n(a|s) q(s,a) = \max_a q(s,a)$. That is, $\pi_n$ converges to a greedy policy with respect to $q$ in a finite number of iterations $n$. $\qquad\square$

### A.12.2  Modified Induction for Bounded Greedification

We modify the induction of the proof of Theorem 3 with finite-sufficient greedification operators, that converge to an $\arg\max$ policy in a finite number of iterations.

*Proof.* **Modified Induction Hypothesis:** There exist $M_{t+1}$ such that for all $m \geq M_{t+1}$ we have $q_{t+1}^m = q_{t+1}^*$, and $\pi_{t+1}^m \cdot q_{t+1}^m = \max_a q_{t+1}^*$.

**Modified Base Case:** Because the convergence to the $\arg\max$ is in finite time (Lemma 8) there exists $M_{H-1}$ such that:

$$\pi_{H-1}^m \cdot q_{H-1}^m = \max_a q_{H-1}^*$$

for all $m \geq M_{H-1}$. Thus the Modified Induction Hypothesis holds at the base case.

**Modified Case $t < H - 1$ Step (1):** Similarly, for all $m \geq M_{t+1}$ we have:

$$\|q_t^{m+1} - q_t^*\| = \|\mathcal{R} + \gamma \mathcal{P}(\pi_{t+1}^{m+1} \cdot q_{t+1}^m) - \mathcal{R} - \gamma \mathcal{P} \max_a q_{t+1}^*\| \tag{50}$$

$$= \gamma \|\mathcal{P}(\pi_{t+1}^{m+1} \cdot q_{t+1}^m) - \mathcal{P} \max_a q_{t+1}^*\| \tag{51}$$

$$\leq \|\mathcal{P}\| \|\pi_{t+1}^{m+1} \cdot q_{t+1}^m - \max_a q_{t+1}^*\| \tag{52}$$

$$= 0 \tag{53}$$

Since also $\|q_t^{m+1} - q_t^*\| \geq 0$, we have $\|q_t^{m+1} - q_t^*\| = 0$ and $q_t^{m+1} = q_t^*$.

**Step (2):** Pick $M_t \geq M_{t+1}$ such that for all $m \geq M_t$ we have $\|\pi_t^m \cdot q_t^m - \max_a q_t^*\| = 0$ which must exist due to convergence to the argmax in finite time of this operator class. Thus, the Modified Induction Hypothesis holds for all states $t$ if it holds for states $t + 1$. $\qquad\square$

## B  Additional Results

### B.1  Value improvement and over estimation

Explicit value-improvement results in greedier evaluation policies. As such, it should directly increase the value targets compared to no value improvement (demonstrated empirically in Figure 1 center). The same can be expected to happen when the value improvement relies on implicit improvement operators such as the expectile operator. Respectively, any increase to the value targets can be expected to interact with (and more specifically, increase) over-estimation bias.

It is well known that overestimation bias can induce pseudo optimistic exploration. This is because increased overestimation bias is more likely to increase overestimation of unvisited state-actions, thus driving exploration into these unvisited state-actions. For that reason, while it can be detrimental in certain environments (as demonstrated by Fujimoto et al., 2018), it can be beneficial in others.

In Figure 4 we investigate the interaction between implicit improvement and over estimation with the expectile operator, with VI-TD3. Each row presents results for a pair of environments. We plot (i) final averaged evaluation return after 3 million environment interactions vs. $\tau$ (the first and third columns from the left). And (ii) we plot final over estimation bias after 3 million environment interactions vs. $\tau$ (the second and last columns from the left).

Generally as $\tau$ increases over estimation bias increases. On the other hand, in many environments the majority of the performance gain is observed for values of $\tau \leq 0.6$ (for example hopper-hop, humanoid-stand/walk/run), for which none to negligible over estimation bias is observed. This is summarized in Table 1

We conclude that while in this domain overestimation can be beneficial (e.g. fish-swim), benefits of VIAC are not limited to the benefits of overestimation ($\tau \leq 0.6$ in hopper-hop, humanoid-stand/walk/run, for example).

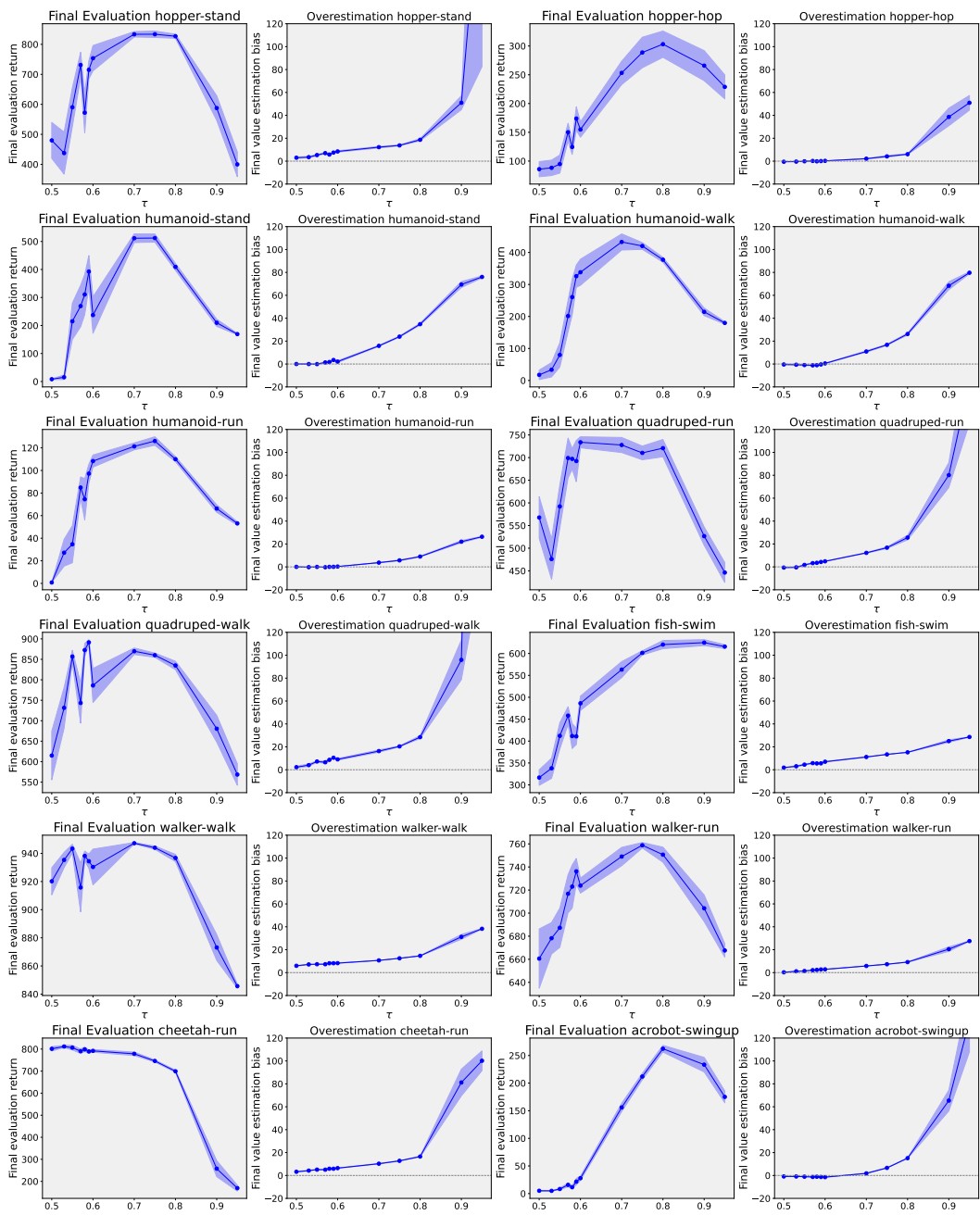

Figure 4: Mean and one standard error across 10 seeds. Left: Final evaluation vs. greedification parameter $\tau$ for VI-TD3 with implicit improvement after $3m$ environment interactions. $\tau = 0.5$ is baseline TD3. Right: Final overestimation bias vs. $\tau$ after $3m$ environment interactions. The majority of the performance increases are independent from an increase in over estimation bias.

Table 1: Performance and overestimation statistics across environments.

| Environment | Performance gain without overest. increase | $\tau$ | Max performance gain | $\tau$ | Overest. as % of max overest. | Overest. as % of performance |
|---|---|---|---|---|---|---|
| hopper-stand | – | – | $353.80 \pm 60.77$ | 0.75 | 5% | 2% |
| hopper-hop | $88.01 \pm 24.31$ | 0.59 | $217.25 \pm 26.38$ | 0.8 | 12% | 2% |
| humanoid-stand | $206.79 \pm 64.93$ | 0.55 | $503.93 \pm 14.72$ | 0.75 | 31% | 5% |
| humanoid-walk | $320.64 \pm 43.42$ | 0.6 | $415.28 \pm 29.29$ | 0.7 | 14% | 3% |
| humanoid-run | $107.50 \pm 5.38$ | 0.6 | $125.25 \pm 3.78$ | 0.75 | 22% | 5% |
| quadruped-run | – | – | $166.18 \pm 47.69$ | 0.6 | 3% | 1% |
| quadruped-walk | – | – | $276.55 \pm 59.85$ | 0.59 | 1% | 1% |
| fish-swim | – | – | $308.07 \pm 19.02$ | 0.9 | 87% | 4% |
| walker-walk | – | – | $27.04 \pm 9.63$ | 0.7 | 28% | 1% |
| walker-run | $17.65 \pm 29.07$ | 0.53 | $98.23 \pm 25.79$ | 0.75 | 27% | 1% |
| cheetah-run | $10.37 \pm 11.18$ | 0.53 | $10.37 \pm 11.18$ | 0.53 | 4% | 1% |
| acrobot-swingup | $22.74 \pm 3.94$ | 0.6 | $257.24 \pm 6.13$ | 0.8 | 11% | 6% |

## B.2   Value Improved TD7

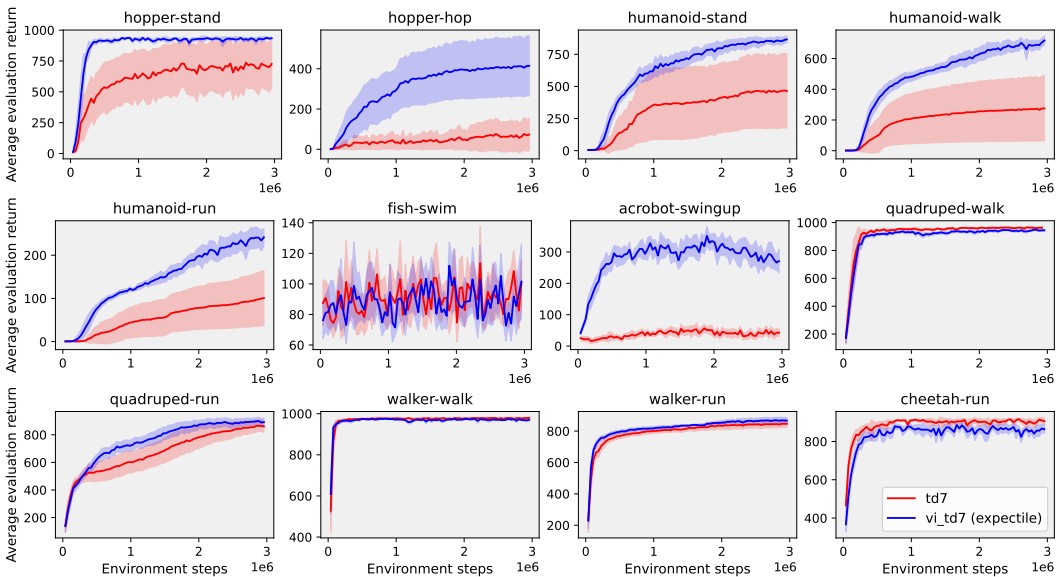

Figure 5: Mean and two standard errors across 10 seeds of VI-TD7 with expectile loss vs. TD7 on the same tasks as Figure 3. Similar performance gains are observed for VI-TD7 in this domain.

## B.3   Increased greedification of the acting policy

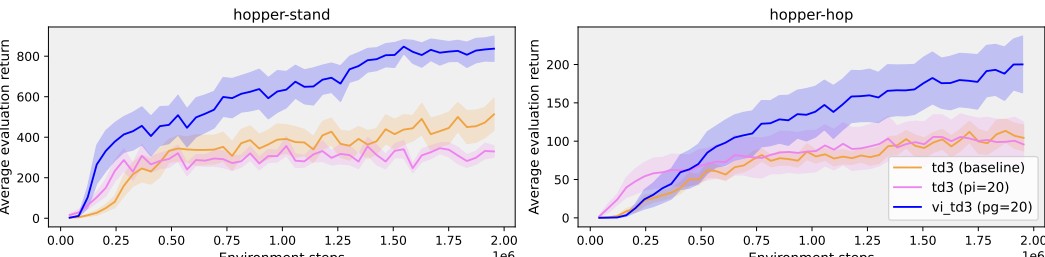

Figure 6: Mean and one standard error across 10 seeds in evaluation of VI-TD3 with Policy Gradient as the value improvement operator, vs. TD3 with 20 repeating policy gradient steps in each update, vs. baseline TD3. Increasing the number of acting-policy updates on the same batch does not contribute to performance.

### B.4 Increased value improvement vs. increased replay ratio

If one is able to spend additional compute on gradient updates, an increased replay ratio is an attractive alternative to value improvement. In Figure 7 we compare VI-TD3 with increasing number of gradient steps to TD3 with increasing replay ratio. In line with similar findings in literature (Chen et al., 2021), replay ratio provides a very strong performance gain for small ratios. As the ratio increases, performance degrades, a result which the literature generally attributes to instability. The VI agent on the other hand does not degrade with increased compute. This suggests a reduced interaction between greedification of the evaluated policy and instability compared to that of the acting policy.

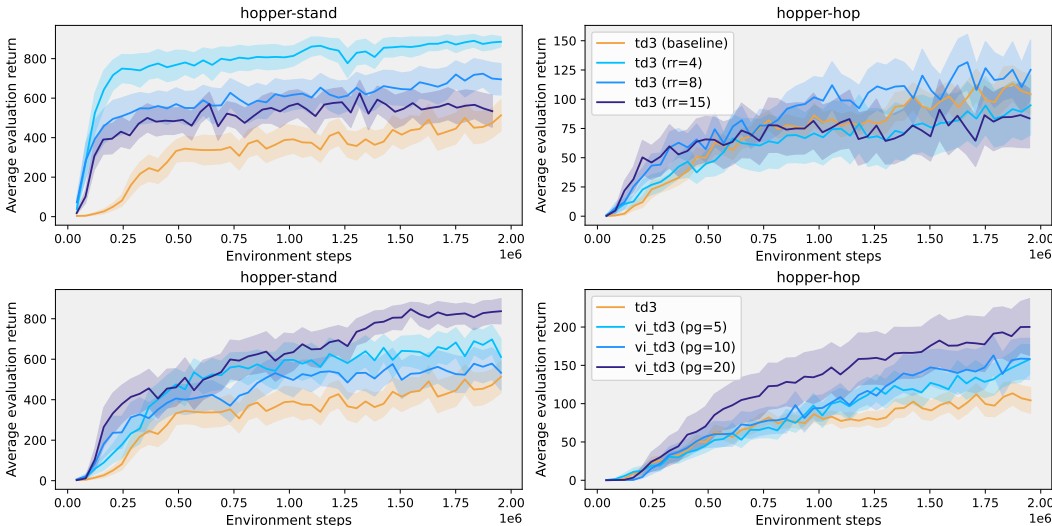

Figure 7: Mean and one standard errors across 10 seeds in evaluation of VI-TD3 with policy-gradient based value improvement vs. td3 with increased replay ratio. Number of gradient steps are equated across rr / VI agent pairs as a pseudo metric for compute. The performance of TD3 generally degrades with increased replay ratio, in line with the results of Chen et al. (2021). In contrast, the performance of VI-TD3 increases with compute, without access to additional mechanisms to address instability.

### B.5 Value Improvement vs. Policy Improvement

A few questions that VIAC naturally raises are *how does the greedification of the acting policy compare to the greedification of the evaluated policy? Is one more important than the other? Does one render the other unnecessary?*

To answer these questions stripped off as many additional influences as possible, we construct a simple experiment in a toy grid environment with Value Improved Generalized Policy Iteration (VIGPI). The VIGPI algorithm uses the operator $\mathcal{I}_{GMZ}$ for both policy improvement (PI) as well as value improvement (VI) with increasing $\beta$. The value table is initialized with 0s. Evaluation uses the Bellman update until $|V_j(s) - V_{j-1}(s)| < 0.0005$ for all states. We plot the value at iteration $i$ of the starting state $s_0$ as a function of $i$.

In the left plot we compare agents with increasing $\beta$ for the *acting policy improvement* (PI). In the right plot we compare agents with increasing $\beta$ for the *evaluated policy improvement* (VI). For both agents we include a final variation which uses the greedy operator for either PI or VI, as well as versions that use $\beta = 0$ (i.e. no greedification).

Since any value improvement that is less greedy than the greedy update depends on the acting policy, slow greedification of the acting policy slows down the convergence of the VI variants. On the other hand, VI + PI is able to significantly increase the rate of convergence compared to the same PI without VI. Similarly, when the rate of greedification $\beta = 0$, with PI there is no learning at all (because the policy never changes) and with VI there is no issue, because it remains a non-detriment operator, and thus does not prevent the improvement of the acting policy (Corollary 2).

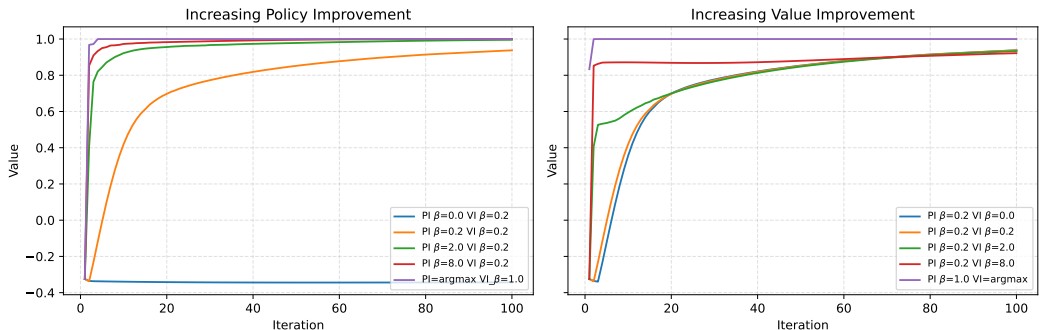

Figure 8: Starting-state value vs. iteration.

## C   Explicit and implicit Value Improved Actor Critic algorithms

In Algorithms 3 and 4 modifications to baseline off-policy Actor Critic are marked in blue.

---

**Algorithm 3** *Explicit* Off-policy Value-Improved Actor Critic

---

1: Initialize policy network $\pi_\theta$, $Q$ network $q_\phi$, Greedification Operators $\mathcal{I}_1$ and $\mathcal{I}_2$, replay buffer $\mathcal{B}$
2: **for** each episode **do**
3:     **for** each environment interaction $t$ **do**
4:         Act $a_t \sim \pi_\theta(s_t)$
5:         Observe $s_{t+1}, r_t$
6:         Add the transition $(s_t, a_t, r_t, s_{t+1})$ to the buffer $\mathcal{B}$
7:         Sample a batch $b$ from $\mathcal{B}$ of transitions of the form $(s_t, a_t, r_t, s_{t+1})$
8:         Update the policy $\pi_\theta(s_t) \leftarrow \mathcal{I}_1(\pi_\theta, q_\phi)(s_t), \forall s_t \in b$
9:         Further improve the policy $\pi'(s_{t+1}) \leftarrow \mathcal{I}_2(\pi_\theta, q_\phi)(s_{t+1}), \forall s_{t+1} \in b$
10:        Sample an action from the improved policy $a \sim \pi'(s_{t+1}), \forall s_{t+1} \in b$
11:        Compute the value targets $y(s_t, a_t) \leftarrow r_t + \gamma q_\phi(s_{t+1}, a), \forall (s_t, a_t, r_t, s_{t+1}) \in b$
12:        Update $q_\phi$ with gradient descent and MSE loss using targets $y$

---

**Algorithm 4** *Implicit* Off-policy Value-Improved Actor Critic

---

1: Initialize policy network $\pi_\theta$, $Q$ network $q_\phi$, Greedification Operator $\mathcal{I}_1$, implicit greedification parameter $\tau$ and replay buffer $\mathcal{B}$
2: **for** each episode **do**
3:     **for** each environment interaction $t$ **do**
4:         Act $a_t \sim \pi_\theta(s_t)$
5:         Observe $s_{t+1}, r_t$
6:         Add the transition $(s_t, a_t, r_t, s_{t+1})$ to the buffer $\mathcal{B}$
7:         Sample a batch $b$ from $\mathcal{B}$ of transitions of the form $(s_t, a_t, r_t, s_{t+1})$
8:         Update the policy $\pi_\theta(s_t) \leftarrow \mathcal{I}_1(\pi_\theta, q_\phi)(s_t), \forall s_t \in b$
9:         Sample an action from the policy $a \sim \pi(s_{t+1}), \forall s_{t+1} \in b$
10:        Compute the value targets $y(s_t, a_t) \leftarrow r_t + \gamma q_\phi(s_{t+1}, a), \forall (s_t, a_t, r_t, s_{t+1}) \in b$
11:        Update $q_\phi$ with gradient descent and $\mathcal{L}_2^\tau$ loss using targets $y$, see Supplement D.2

---

## D   Experimental Details

### D.1   Gradient-Based VI-TD3

Gradient-based VI-TD3 copies the existing policy used to compute value targets (the target policy, in TD3) $\pi_{\theta'}$ into a new policy $\pi'_{\theta'}$. The algorithm executes N repeating gradient steps on $\pi'_{\theta'}$ with respect to states $s_{t+1} \in b$ with the same operator TD3 uses to improve the policy (the deterministic policy gradient) and with respect to the same batch $b$. The value-improved target $y(s_t, a_t)$ is computed in the

same manner to the original target of TD3 but with the fresh greedified target network $\pi'_{\theta'}$. In TD3, that summarizes to sampling an action from a clipped Gaussian distribution with mean $\pi'_{\theta'}(s_{t+1})$, variance parameter $\sigma$ and clipped between $(-\beta, \beta)$:

$$a' \sim \mathcal{N}(\pi'_{\theta'}(s_{t+1}), \sigma).clip(-\beta, \beta) \tag{54}$$

And using the action $a'$ to compute the value target in the Sarsa manner:

$$y(s_t, a_t) = r_t + \gamma \min_{i \in \{1,2\}} q_{\phi_i}(s_{t+1}, a'), \forall (s_t, a_t, r_t, s_{t+1}) \in b \tag{55}$$

The policy used to compute the value targets $\pi_{\theta'}$ is then discarded.

### D.2 Implicit Policy Improvement with Expectile Loss

The expectile-loss $\mathcal{L}_2^\tau$ proposed by Kostrikov et al. (2022) as an implicit policy improvement operator for continuous-domain Q-learning can be formulated as follows: when $y(s_t, a_t) > q(s_t, a_t)$ (the target is greater than the prediction), the loss equals $\tau(y(s_t, a_t) - q(s_t, a_t))^2$. When $y(s_t, a_t) \leq q(s_t, a_t)$ (the target is smaller than the prediction) the loss equals $(1 - \tau)(y(s_t, a_t) - q(s_t, a_t))^2$. If $\tau = 0.5$, this loss is equivalent to the baseline $\mathcal{L}_2$ loss. Intuitively, when $\tau > 0.5$ the agent favors errors where the prediction should increase, over predictions where it should reduce. I.e. the agent favors targets where $\pi'(s_{t+1})$ (the implicit policy evaluated on the next state) chooses "better" actions than the current policy, directly approximating the value of an improved policy.

By imposing this loss on the value network, in stochastic environments the network may learn to be *risk-seeking*, by implicitly favoring interactions $s_t, a_t, r_t, s_{t+1}$ where the observed $r_t$ was large or the state $s_{t+1}$ was favorable. This is addressed by Kostrikov et al. (2022) by learning an additional $v_\psi$ network that is trained with the expectile loss, while the $q$ network is trained with SARSA targets $r_t + \gamma v_\psi(s_{t+1})$ and the regular $\mathcal{L}_2$ loss, while the $v_\psi$ network is trained with targets $y(s_t, a_t) = q_\phi(s_t, a_t)$ and the expectile loss. In deterministic environments this is not necessary however, and in our experiments we have directly replaced the $\mathcal{L}_2$ loss on the value $q_\phi$ with the expectile loss.

The value target $y(s_t, a_t)$ remained the unmodified target used by TD3 / SAC respectively.

Another aspect where our use of the expectile loss diverges from that of Kostrikov et al. (2022), is that we use it with respect to the *online acting policy* $\pi_\theta$, rather than with respect to the policy captured by the replay buffer. That is, the original targets of IQL are:

$$y(s_t, a_t) = r_t + \gamma v_\psi(s_{t+1}), \quad (s_t, a_t, r_t, s_{t+1}) \in \mathcal{D}. \tag{56}$$

On the other hand, our usage is with respect to $\pi_\theta$:

$$y(s_t, a_t) = r_t + \gamma q_\phi(s_{t+1}, \pi_\theta(s_{t+1})), \quad (s_t, r_t, s_{t+1}) \in \mathcal{D}. \tag{57}$$

This results in targets that are much more on policy (more "fresh", if you will), compared to the information in the replay buffer, unless the replay buffer contains trajectories exclusively from $\pi_\theta$. This also enables compute more accurate targets by averaging across any number of fresh samples $\{a_i\}_{i=1}^N \sim \pi_\theta(s_{t+1})$ with: $y(s_t, a_t) = r_t + \gamma \frac{1}{N} \sum_{i=1}^N q_\phi(s_{t+1}, a_i)$, although this was not used in our experiments.

### D.3 Evaluation Method

We plot the mean and standard error for *evaluation curves* across multiple seeds. Evaluation curves are computed as follows: after every $n = 5000$ interactions with the environment, $m = 3$ evaluation episodes are ran with the latest network of the agent (actor and critic). The score of the agent is the return averaged across the $m$ episodes. The actions in evaluation are chosen deterministically for TD3, SAC and TD7 with the mean of the policy (the agents use Gaussian policies). The evaluation episodes are not included in the agent's replay buffer or used for training, nor do they count towards the number of interactions.

### D.4 Compute

The experiments were run on the internal compute cluster Delft AI Cluster (DAIC) (2024) using any of the following GPU architectures: NVIDIA Quadro K2200, Tesla P100, GeForce GTX 1080 Ti, GeForce RTX 2080 Ti, Tesla V100S and Nvidia A-40. Each seed was ran on one GPU, and was

given access to 6GB of RAM and 2 CPU cores. Total training wall-clock time averages were in the range of $0.5$ to $2$ hours per $10^6$ environment steps, depending on GPU architecture, the baseline algorithm and VI variations. For example, baseline TD3 wall-clock time averages were roughly $1.25$ hours per $10^6$ environment steps on average. The total wall clock time over all experiments presented in this paper (main results, baselines and ablations) is estimated at $\approx 12000$ wall-clock hours of the compute resources detailed above: $\approx 7320$ for the results in the paper and $\approx 4300$ for the ablations in the appendix. Additional experiments that are not included in the paper were run in the process of implementation and testing.

## D.5  Implementation & Hyperparemeter Tuning

Our implementation of TD3 and SAC relies on the popular code base CleanRL (Huang et al., 2022). CleanRL consists of implementations of many popular RL algorithms which are carefully tuned to match or improve upon the performance reported in the original paper. The implementations of TD3 and SAC use the same hyperparameters as used by the authors (Fujimoto et al. (2018) and Haarnoja et al. (2018a) respectively), with the exception of the different learning rates for the actor and the critic in SAC, which were tuned by CleanRL.

For the TD7 agent, we use the original implementation by the authors (Fujimoto et al., 2023), adapting the action space to the DeepMind control's in the same manner as CleanRL's implementation of TD3. Additionally, a non-prioritized replay buffer has been used for TD7 which was used by the TD3 and SAC agents as well. The hyperparameters are the same as used by the author.

The VI-variations of all algorithms use the same hyperparameters as the baseline algorithms without any additional tuning, with the exception of grid search for the greedification parameters $\tau$ presented in Figure 2.

## D.6  Network Architectures

The experiments presented in this paper rely on standard architectures for every baseline. TD3 and SAC used the same architecture, with the exception that SAC's policy network predicts a mean of a Gaussian distribution as well as standard deviation, while TD3 predicts only the mean. TD7 used the same architecture proposed and used by Fujimoto et al. (2023).

**TD3 and SAC**:

Actor: 3 layer MLP of width 256 per layer, with ReLU activations on the hidden layers. The final action prediction is passed through a `tanh` function.

Critic: 3 layer MLP of width 256 per layer, with ReLU activations on the hidden layers and no activation on the output layer.

**TD7**: Has a more complex architecture, which is specified in (Fujimoto et al., 2023).

## D.7  Hyperparemeters

| TD3 | | SAC | | TD7 | |
|---|---|---|---|---|---|
| exploration noise | 0.1 | | | exploration noise | 0.1 |
| Target policy noise | 0.2 | | | Target policy noise | 0.2 |
| Target smoothing | 0.005 | Target smoothing | 0.005 | | |
| noise clip | 0.5 | auto tuning of entropy | True | noise clip | 0.5 |
| | | Critic learning rate | 1e-3 | Critic learning rate | 3e-4 |
| Learning rate | 3e-4 | Policy learning rate | 3e-4 | Policy learning rate | 3e-4 |
| Policy update frequency | 2 | Policy update frequency | 2 | Policy update frequency | 2 |
| $\gamma$ | 0.99 | $\gamma$ | 0.99 | $\gamma$ | 0.99 |
| Buffer size | $10^6$ | Buffer size | $10^6$ | Buffer size | $10^6$ |
| Batch size | 256 | Batch size | 256 | Batch size | 256 |
| learning start | $10^4$ | learning start | $10^4$ | learning start | $10^4$ |
| evaluation frequency | 5000 | evaluation frequency | 5000 | evaluation frequency | 5000 |
| Num. eval. episodes | 3 | Num. eval. episodes | 3 | Num. eval. episodes | 3 |

