# OpenReview forum: "Value Improved Actor Critic Algorithms"
_NeurIPS.cc/2025/Conference — NeurIPS 2025 poster_

### Official Review · Reviewer_y6Pz · 2025-06-20

**Clarity:** 4
**Significance:** 3
**Originality:** 3
**Rating:** 5
**Confidence:** 3

**Summary:**

This paper studies Value Improved Actor Critic (VAIC) to address the tradeoff between fast improvement (greedification) and stability. The authors propose a novel framework that decouples the acting policy from the evaluated policy in actor-critic. This allows for greedier value improvements to the critic, while keeping stable updates for the actor. Empirically, the proposed VAIC variants of TD3 and SAC outperform or match the original methods across several DeepMind control suite tasks.

**Questions:**

In explicit VI-TD3, does the extra greediness (e.g., more PG steps) lead to an increase in total compute/wall-clock time, even if environment step efficiency is improved?

**Ethical Concerns:**

["NO or VERY MINOR ethics concerns only"]

**Final Justification:**

I do not have deep expertise in this topic, so please defer to any more expert reviewer than me.
However, from reading all the reviews and author discussion, I don't see any fatal flaws, and there seem to be several good theoretical insights and practical algorithmic contributions coming out.
So I recommend accept.

**Limitations:**

Yes.

**Paper Formatting Concerns:**

None.

**Quality:**

3

**Strengths And Weaknesses:**

Strengths:

+	Theoretical contribution about convergence in actor critic is new and insightful, AFAIK.  Has necessary and sufficient conditions to guarantee that policy iteration converges to optimal policy. Shows that Policy improvement is not a sufficient condition for the convergence of Policy Iteration.

+	Practical algorithmic suggestion about decoupled acting and evaluated policy.

+	Concrete implementation with explicit and implicit greedification strategies, with associated positive empirical results on VAIC for TD3 and SAC.

+	Generally clearly written paper and insightful ablation and analysis experiments. (Analysis around Fig4-7 are all appreciated).

Weaknesses:

-	Analysis seems to be limited to finite-horizon MDPs.

-	Limited empirical study on stability-greediness tradeoff. This is a keystone of the paper’s narrative, but there is only limited empirical analysis of this.

-	Empirical validation limited to  low-dimensional continuous control, not discrete action or high-dimensional inputs.

-	Introduces a new greedification hyperparameter to tune in RL.

---

> ### Author Rebuttal · Authors · 2025-07-28
>
> Dear reviewer,
>
> Thank you for your valuable time, thoughtful review and accurate comments.
>
> **2. Limited empirical study on stability-greediness tradeoff. This is a keystone of the paper’s narrative, but there is only limited empirical analysis of this.**
>
> We agree with the reviewer that the tradeoff can be investigated further.
> We note that aspects of it have been investigated in prior work (for example [1], beyond the works already cited in the paper), and that different faces of it are investigated in Figures 1, 2, 4, 6 and 7.
>
> **4. Introduces a new greedification hyperparameter to tune in RL.**
>
> One could say that the tradeoff was there before VIAC, and with it being there, there was a possibility to tune it.
> However, we are of course in agreement with the reviewer that the introduction of any additional hyperparameter is unfortunate.
> We will include in the appendix a figure like Figure 4, extended for all environments, that sheds more light on the effects of different $\tau$.
>
> **Q1: In explicit VI-TD3, does the extra greediness (e.g., more PG steps) lead to an increase in total compute/wall-clock time, even if environment step efficiency is improved?**
>
> *TLDR: Yes, absolutely. However, it is possible to get signficant (stronger, in fact) performance improvement with much smaller compute cost than PG with other explicit greedification operators. Results for which are included below.*
>
> Yes indeed, increasing PG steps comes with an increase in compute cost, which is important to mention (included in line 299). However, it is also possible to see significant performance improvement with explicit greedification of the evaluated policy without insignificant compute increase:
>
> A simple example of a value improvement operator that is inexpensive and shows strong performance benefits in these domains is best-of-n (BON): sample N actions, take the value of the maximizing action as the bootstrap for the value target.
> This is a popular operator in RL for language models [2].
> We include a table that summarizes final performance for VI-TD3 with this operator across the same domains as in Figure 3.
> We include baseline TD3, VI-TD3 with the expectile operator ($\tau=0.75$) and VI-TD3 with BON ($N=64$).
>
> BON shows similar performance gains to IQL across most domains.
>
> Below we report final-performance results normalized per environment between minimum and maximum final return over all seeds (20 for TD3 and VI-TD3 (IQL), and 12 for VI-TD3 (BON)), averaged per agent over environments, with +- one SEM.
> The IQL and BON variations are within each others 2 SEM confidence bound.
>
> |Agent|Normalized Final Perf.|
> |-|-|
> |Baseline TD3|0.41 ± 0.05|
> |VI-TD3 (IQL)|0.75 ± 0.04|
> |VI-TD3 (BON)|0.68 ± 0.04|
>
> Below are the unnormalized final-performance results per environment along with 1 SEM which show the similarities in performance gains across the majority of environments in more detail.
>
> |Environment|Final perf. TD3|Final perf. VI-TD3 (expectile)|Final perf. VI-TD3 (BON)|
> |-|-|-|-|
> |hopper-stand|670.9 ± 87.5|**936.4** ± 5.1|**900.4** ± 24.4|
> |hopper-hop|135.8 ± 29.6|**377.6** ± 34.0|250.2 ± 38.9|
> |humanoid-stand|26.4 ± 17.8|666.6 ± 15.2|**723.1** ± 24.9|
> |humanoid-walk|38.8 ± 34.5|**540.5** ± 12.5|**560.5** ± 52.4|
> |humanoid-run|0.9 ± 0.1|**172.4** ± 6.3|**180.7** ± 7.3|
> |fish-swim|489.9 ± 74.6|**753.2** ± 21.6|492.9 ± 60.1|
> |acrobot-swingup|9.4 ± 3.3|**317.2** ± 36.6|5.4 ± 4.3|
> |quadruped-walk|706.9 ± 86.1|**923**.6 ± 9.7|**943.1** ± 5.3|
> |quadruped-run|**764.3** ± 51.6|**838.6** ± 19.5|**863.1** ± 30.3|
> |walker-walk|**971.0** ± 2.0|**966.3** ± 3.6|**974.9** ± 2.7|
> |walker-run|792.4 ± 16.0|**852.4** ± 10.5|**833.8** ± 5.0|
> |cheetah-run|**897.3** ± 4.4|**831.3** ± 24.5|**848.9** ± 27.4|
>
> [1] Thrun, Sebastian, and Anton Schwartz. "Issues in using function approximation for reinforcement learning." Proceedings of the 1993 connectionist models summer school. Psychology Press, 2014.
>
> [2] Huang, Audrey, et al. "Is best-of-n the best of them? coverage, scaling, and optimality in inference-time alignment." arXiv preprint arXiv:2503.21878 (2025).

---

### Official Review · Reviewer_SmgB · 2025-06-27

**Clarity:** 1
**Significance:** 2
**Originality:** 3
**Rating:** 4
**Confidence:** 3

**Summary:**

This paper introduces Value Improved Actor-Critic, a general technique for AC algorithms to balance the greediness and stability during training. The key idea behind is to follow the value improvement -> policy improvement paradigm. Moreover, the authors choose to apply slower policy improvement for the actual policy for stability reason. The method is experimented on both TD-3 and SAC on dm-control benchmark, verifying that VIAC can consistently improves the performance and has the capability to adjust the tradeoff via simple hyper-parameter tuning.

**Questions:**

- Is there any other greedy operators that also works well and where is the experiment for Gumbel MuZero operator? Have the author explored this direction?

**Ethical Concerns:**

["NO or VERY MINOR ethics concerns only"]

**Final Justification:**

The authors have clarified the policy stability claim and provided a detailed explanation on potential operators with a comprehensive evaluation.

**Limitations:**

I didn't find any place the author discussed the limitation.

**Quality:**

3

**Strengths And Weaknesses:**

Strength:
- Well-motivated to solve the trade-off between greediness and stability.
- Has strong theoretical foundation.
- Proposes explicit and implicit methods, and well addresses the computational overhead for repetitive gradient update.
- Generality is well-supported by both deterministic and stochastic policies.

Weakness:
- The presentation has a lot of room to be improved. Too much space is spent for the justification of generalized policy iteration. The authors should clearly explain the motivation and connection to AC algorithms at first, otherwise it is hard to follow.
- The authors claim that the slower policy improvement brings stability, which I believe is also a key difference between the method and other off-policy algorithms that improve UTD ratios. However, there is no ablations to show the effectiveness of this design choice.
- There is no proofs of convergence or theoretical guarantees of improvement over standard actor-critic for both deterministic and stochastic policies. Compared to the dense proofs for Section 3, this looks weird.
- For experiments in dm-control suite, the highest rewards the method achieved are still far from sota performance. For example, for humanoid stand and run, sota algorithms can reach 1000~ and 600~ [1,2], which are almost doubled compared to the proposed method. I note this is a minor issue.

[1] Nauman, Michal, et al. "Bigger, regularized, optimistic: scaling for compute and sample-efficient continuous control." arXiv preprint arXiv:2405.16158 (2024).

[2] Celik, Onur, et al. "Dime: Diffusion-based maximum entropy reinforcement learning." arXiv preprint arXiv:2502.02316 (2025).

---

> ### Author Rebuttal · Authors · 2025-07-28
>
> Dear reviewer,
>
> Thank you for your valuable time and clear comments and questions.
>
> **1. The presentation has a lot of room to be improved. Too much space is spent for the justification of generalized policy iteration. The authors should clearly explain the motivation and connection to AC algorithms at first, otherwise it is hard to follow.**
>
> Thanks for pointing that out, we will make sure that motivation and connection to ACs is made earlier and clearer.
>
> **2. The authors claim that the slower policy improvement brings stability, which we believe is also a key difference between the method and other off-policy algorithms that improve UTD ratios. However, there is no ablations to show the effectiveness of this design choice.**
>
> The claim that the slower policy improvement can bring stability is motivated by prior work:
> [6] explicitly delay the policy updates for learning stability. [7] and [8] explicitly limit the maximum change allowed to the policy before each new data-gathering step.
> [9] explicitly refers to and addresses issues that rise from too-fast changes to the policy.
>
> In regards specifically to VIAC, Figure 7 demonstrates that increased UTD destabilizes the learning process much earlier than the increased greedification of the evaluated policy, equated over the same budget of gradient steps.
>
> **3. There is no proofs of convergence or theoretical guarantees of improvement over standard actor-critic for both deterministic and stochastic policies. Compared to the dense proofs for Section 3, this looks weird.**
>
> *In regards to convergence:*
> Using DP to analyze convergence properties of AC algorithms is rather standard [1, 2].
> The main advantage of this analysis, from our perspective, is in its generality: the results are not limited to specific improvement operators, gradient based updates, presence of local optima, etc. and thus can be used to underly the convergence of algorithms that go beyong AC, such as MuZero [10].
>
> *In regards to improvement over standard actor-critics (or even policy iteration):*
> Under strict-greedification ($>$) assumptions, we would expect that it is possible to show that value-improved GPI can outperform GPI for specific combinations of operators.
> Since a small number of recent algorithms already fall within the framework of VIAC in practice, albeit without the perspective of the framework (see lines 323 to 336 in the related work), we viewed proving soundness of this framework as more important than proving the possibility of benefit (which from that perspective, has already been observed).
> However, we agree that this result is important as well and we hope it will be addressed in future work.
>
> **4. For experiments in dm-control suite, the highest rewards the method achieved are still far from sota performance. For example, for humanoid stand and run, sota algorithms can reach 1000~ and 600~ [1,2], which are almost doubled compared to the proposed method. I note this is a minor issue.**
>
> BRO [3] specifically focuses on scaling up the DNN architectures and pairing the agent with dedicated optimism-based exploration, while Dime [4] explicitly changes the policy class and DNN architectures. In contrast, the baseline TD3 and SAC over which we investigate the effect of VI do not use any of these advancements. However, the results in Figure 5 in the appendix (VI-TD7) show that the improvements of VIAC are not limited to the older architectures of TD3 and SAC, and indeed VI-TD7 results in better performance across several environments compared to VI-TD3 / VI-SAC on both humanoid-stand and humanoid-run, suggesting that the improvement from VIAC is orthogonal to improvement from DNN architectures.
>
> **Q1. Is there any other greedy operators that also works well and where is the experiment for Gumbel MuZero operator? Have the author explored this direction?**
>
> *TLDR: yes, and they have also showed significant performance gains in this domain.*
>
> We have tried several explicit greedification operators, in addition to PG and the implicit IQL. Overall, all resulted in performance gains in this domain.
> We include best of N (BON) as an example:
> BON samples $N$ actions and takes the value of the maximizing action as the bootstrap for the value target, and is popular in RL for language models [5].
> The main advantage of IQL in comparison to the explicit operators is not only in its minimal computational and implementation cost but also that the greedification can be tuned with $\tau$ without increasing cost. In contrast, in operators such as BON, increasing greedification translates to increasing $N$, where the compute cost of the operator also increases with $N$.
>
> In the experiments reported below, BON shows similar performance gains to IQL across most domains.
>
> We first report final-performance results normalized per environment between minimum and maximum final return over all seeds (20 for TD3 and VI-TD3 (IQL), and 12 for VI-TD3 (BON)), averaged per agent over environments, with +- one SEM.
> For BON, we use $N=64$, and for IQL we use the same $\tau=0.75$.
> The IQL and BON variations are within each others 2 SEM confidence bound.
>
> |Agent|Normalized Final Perf.|
> |-|-|
> |Baseline TD3|0.41 ± 0.05|
> |VI-TD3 (IQL)|0.75 ± 0.04|
> |VI-TD3 (BON)|0.68 ± 0.04|
>
> Below are the unnormalized final-performance results per environment along with 1 SEM which show the similarities in performance gains across the majority of environments in more detail.
>
> |Environment|Final perf. TD3|Final perf. VI-TD3 (expectile)|Final perf. VI-TD3 (BON)|
> |-|-|-|-|
> |hopper-stand|670.9 ± 87.5|**936.4** ± 5.1|**900.4** ± 24.4|
> |hopper-hop|135.8 ± 29.6|**377.6** ± 34.0|250.2 ± 38.9|
> |humanoid-stand|26.4 ± 17.8|666.6 ± 15.2|**723.1** ± 24.9|
> |humanoid-walk|38.8 ± 34.5|**540.5** ± 12.5|**560.5** ± 52.4|
> |humanoid-run|0.9 ± 0.1|**172.4** ± 6.3|**180.7** ± 7.3|
> |fish-swim|489.9 ± 74.6|**753.2** ± 21.6|492.9 ± 60.1|
> |acrobot-swingup|9.4 ± 3.3|**317.2** ± 36.6|5.4 ± 4.3|
> |quadruped-walk|706.9 ± 86.1|**923**.6 ± 9.7|**943.1** ± 5.3|
> |quadruped-run|**764.3** ± 51.6|**838.6** ± 19.5|**863.1** ± 30.3|
> |walker-walk|**971.0** ± 2.0|**966.3** ± 3.6|**974.9** ± 2.7|
> |walker-run|792.4 ± 16.0|**852.4** ± 10.5|**833.8** ± 5.0|
> |cheetah-run|**897.3** ± 4.4|**831.3** ± 24.5|**848.9** ± 27.4|
>
> We will include these results in the appendix.
>
> **I didn't find any place the author discussed the limitation.**
>
> The main limitation we observe for our contribution is the limitations of the theoretical results (finite horizon MDPs, DP, etc) which we mention per theorem. We will make sure this is made more clear and explicit in the text.
>
> *Please let us know if there remain any unaddressed concerns, comments or questions.*
>
> [1] Tsitsiklis, John N. "On the convergence of optimistic policy iteration." Journal of Machine Learning Research 3.Jul (2002): 59-72.
>
> [2] Smirnova, Elena, and Elvis Dohmatob. "On the convergence of approximate and regularized policy iteration schemes." arXiv preprint arXiv:1909.09621 (2019).
>
> [3] Nauman, Michal, et al. "Bigger, regularized, optimistic: scaling for compute and sample-efficient continuous control." arXiv preprint arXiv:2405.16158 (2024).
>
> [4] Celik, Onur, et al. "Dime: Diffusion-based maximum entropy reinforcement learning." arXiv preprint arXiv:2502.02316 (2025).
>
> [5] Huang, Audrey, et al. "Is best-of-n the best of them? coverage, scaling, and optimality in inference-time alignment." arXiv preprint arXiv:2503.21878 (2025).
>
> [6] Fujimoto, Scott, Herke Hoof, and David Meger. "Addressing function approximation error in actor-critic methods." International conference on machine learning. PMLR, 2018.
>
> [7] Schulman, John, et al. "Trust region policy optimization." International conference on machine learning. PMLR, 2015.
>
> [8] Schulman, John, et al. "Proximal policy optimization algorithms." arXiv preprint arXiv:1707.06347 (2017).
>
> [9] Capone, Cristiano, and Paolo Muratore. "Learning fast while changing slow in spiking neural networks." Neuromorphic Computing and Engineering 4.3 (2024): 034002.
>
> [10] Danihelka, Ivo, et al. "Policy improvement by planning with Gumbel." International Conference on Learning Representations. 2022.

---

> > ### Comment · Reviewer_SmgB · 2025-08-02
> >
> > I have read the detailed rebuttal and my concerns are addressed. I recommend the authors to include those changes and clarifications in the revised version and will raise my score accordingly.

---

> > > ### Author Response · Authors · 2025-08-03
> > >
> > > We thank the reviewer for acknowledging our rebuttal. We will include the requested changes and clarifications, as well as all additional results in the revised version of the paper.

---

### Official Review · Reviewer_HLEH · 2025-06-29

**Clarity:** 2
**Significance:** 3
**Originality:** 3
**Rating:** 5
**Confidence:** 3

**Summary:**

The paper studies a class of actor critic algorithms that separates between an "evaluated policy" and the "acting policy". The evaluated policy is induced by a greedification operator that is separate from the greedification operator of the acting policy.
As a result, the policy evaluation phase effectively generates additional "value improvement" by means of evaluating an improved policy.
These algorithms are called Value Improved Actor Critic Algorithms (VIAC).

The core idea is that value improvement can be controlled separately from the the greedification of the acting policy, which may be constrained to support greedification operators that are more limited than those of the value improvement.

The paper studies this class of algorithms within the generalized policy iteration framework, and presents elementary convergence results. Further, experiments are carried that indicate VIAC offer clear improvement over non value improved baselines TD3, TD7, and SAC on continuous control tasks.

**Questions:**

- Line 247: Typo "the sets are not also not disjoint"
- Is $I_{gmz}$ just a slight generalization (by use of $\sigma$) of the standard soft policy iteration greedification operator?
- The notion of "non-detriment" operator is first introduced in the statement of Corollary 2, and it's meaning is unclear (the "e.g." just points back to the definition of a greedification operator).
- Lines 282-283: It is unclear what is the difference between questions 1 and 2.
- It is unclear how you map the greedification operators in Alg2 to those of Alg3. When you increase the $n$ of $pg=n$, do you increase it for both the evaluated policy and acting policy?

**Ethical Concerns:**

["NO or VERY MINOR ethics concerns only"]

**Final Justification:**

Following the authors detailed rebuttal and discussions that followed, most of my concerns have been addressed. In particular, the authors have convinced me the framework through which they study VIAC makes sense, and that the performance gains reported in the paper originate from the combination of the two (value / acting policy) improvement mechanisms.
As a result, I choose to raise my rating 4 --> 5, favoring clear acceptance.

**Limitations:**

Yes

**Quality:**

3

**Strengths And Weaknesses:**

### Strengths
* The paper makes an appealing observation, that value improvement and policy improvement may be subject to different considerations, and thus it is worthwhile exploring approaches that decouple them.
* Experiments indicate the approach has merit, improving over existing baselines.

### Weaknesses
* It is ultimately unclear to me where the performance gain really comes from. Is there really an interaction of the two improvement operators, or is this algorithm more of an online version of IQL? See comments 1+2 below.
* Much of the paper is dedicated to introduction of the theory, that ultimately does not reveal much other than sort of a sanity check with very limited scope. (While I consider this beneficial to some extent, it is currently occupying quite a bit of real estate considering its added value.)

Comments:
1. I was left uncertain as to whether the generalized policy iteration is the right framework to study the VIAC algorithm in its implicit value improvement flavor. Is the greedification of the acting policy important in any way, or is the implicit value improvement really doing all the work, in which case the algorithm is question is an online (offpolicy) version of IQL? In particular, perhaps the experiments involving implicit value improvement (i.e., Figure 3), would benefit from another IQL-like baseline, where the improvement of the acting policy is ablated in some way.
2. The presentation of Figure 1 with x axis being the environment steps is questionable, given that the compute would be the right reference to use ofor the comparison.

---

> ### Author Rebuttal · Authors · 2025-07-28
>
> Dear reviewer,
>
> Thank you for your valuable time, detailed review and many thoughtful comments and questions.
>
> **1. It is ultimately unclear to me where the performance gain really comes from. Is the greedification of the acting policy important in any way, or is the implicit value improvement really doing all the work, in which case the algorithm is question is an online (offpolicy) version of IQL?**
>
> *TLDR: experiments with other value-improvement operators show that signif. performance gains from VI are not unique to IQL. We conclude that VIAC is not simply online IQL.*
>
> Best-of-n (BON) is an explicit value improvement operator which samples N actions and takes the value of the maximizing action as the bootstrap for the value target.
> BON is a popular operator in RL for language models [1].
> We include a table that summarizes the final performance for baseline TD3, and the IQL and BON variations of VI-TD3, where BON shows similar performance gains to IQL across most domains.
> For BON, we use $N=64$, and for IQL we use the same $\tau=0.75$.
> Normalized per environment between minimum and maximum final return over all seeds (20 for TD3 and VI-TD3 (IQL), and 12 for VI-TD3 (BON)), averaged per agent over environments, with +- one SEM.
> The IQL and BON variations are within each others 2 SEM confidence bound.
>
> |Agent|Normalized Final Perf.|
> |-|-|
> |Baseline TD3|0.41 ± 0.05|
> |VI-TD3 (IQL)|0.75 ± 0.04|
> |VI-TD3 (BON)|0.68 ± 0.04|
>
> Below are the unnormalized final-performance results per environment along with 1 SEM which show the similarities in performance gains across the majority of environments in more detail.
>
> |Environment|Final perf. TD3|Final perf. VI-TD3 (expectile)|Final perf. VI-TD3 (BON)|
> |-|-|-|-|
> |hopper-stand|670.9 ± 87.5|**936.4** ± 5.1|**900.4** ± 24.4|
> |hopper-hop|135.8 ± 29.6|**377.6** ± 34.0|250.2 ± 38.9|
> |humanoid-stand|26.4 ± 17.8|666.6 ± 15.2|**723.1** ± 24.9|
> |humanoid-walk|38.8 ± 34.5|**540.5** ± 12.5|**560.5** ± 52.4|
> |humanoid-run|0.9 ± 0.1|**172.4** ± 6.3|**180.7** ± 7.3|
> |fish-swim|489.9 ± 74.6|**753.2** ± 21.6|492.9 ± 60.1|
> |acrobot-swingup|9.4 ± 3.3|**317.2** ± 36.6|5.4 ± 4.3|
> |quadruped-walk|706.9 ± 86.1|**923**.6 ± 9.7|**943.1** ± 5.3|
> |quadruped-run|**764.3** ± 51.6|**838.6** ± 19.5|**863.1** ± 30.3|
> |walker-walk|**971.0** ± 2.0|**966.3** ± 3.6|**974.9** ± 2.7|
> |walker-run|792.4 ± 16.0|**852.4** ± 10.5|**833.8** ± 5.0|
> |cheetah-run|**897.3** ± 4.4|**831.3** ± 24.5|**848.9** ± 27.4|
>
> We will include these results in the appendix.
>
> **2. Much of the paper is dedicated to introduction of the theory, that ultimately does not reveal much other than sort of a sanity check with very limited scope.**
>
> Combining value improvement with policy improvement is already done in practice (see lines 326 to 336), albeit generally without this perspective in mind.
> For that reason, we think that the sanity check is important - if this approach is not generally sound we should not or should be careful when we use it.
> The choice of very abstract, very general scope of GPI allows us to establish Theorem 2, policy improvement is not enough, and the nec. / suff. conditions of Definitions 3-5, under the most abstract conditions, that are not limited to function approximation or even to RL.
> Analyzing VIAC in this scope allows us to further establish that this novel algorithms approach to RL/DP is sound under the most general conditions (excepting for finite horizon).
> We believe that it is an important first step towards establishing the theoretical understanding that underpins this general perspective which diverges from the standard RL iteration of policy improvement / policy evaluation.
>
> **C1. I was left uncertain as to whether the generalized policy iteration is the right framework to study the VIAC algorithm in its implicit value improvement flavor.**
>
> *TLDR: We conclude that: 1. implicit improvement is within VIAC (including specifically the way we use IQL), and 2. that the value and policy improvement compliment each other, and unless one of them is explicitly the argmax, they do not render each other obselete.*
>
> If we understand correctly (please correct us otherwise), the concern is that the improvement in the value is doing all the work, and the improvement in the policy becomes unnecessary / unimportant.
>
> We first note that if the greedy operator is used the VI-GPI collapses to Value Iteration / Policy Iteration (which are the same algorithm, from the perspective of GPI). In which case, the value improvement and the policy improvemnt coincide.
> However, as long as both operators take $\pi$ into account (for example, operators like $I_{GMZ}$ or policy gradient), every improvement step depends strongly on $\pi$. If $I(\pi, Q)$ does not diverge far from $\pi$ the value-improved value function does evaluate policies that are very different from $\pi$.
>
> In fact, in a very simple set of experiments with dynamic programming and VI-GPI, discrete actions, and the same operators for both value and policy improvement ($I_{GMZ}$), we found that the improvement to the policy is the dominating effect.
> If this experiment sounds interesting despite its simplicity, please let us know and we will include it.
>
> Specifically in relation to the implicit improvement of the expectile - note that we use it with respect to freshly sampled actions from the acting policy $\pi_\theta$, not with respect to actions stored in the replay buffer. [2] prove that this evaluates an explicitly greedier policy with respect to $\pi_\theta$, and [3] show that this policy is real and can be extracted.
>
> Please let us know if the comment was not understood correctly, and this does not address / answer the concern the reviewer has in mind.
>
> **C2. The presentation of Figure 1 with x axis being the environment steps is questionable, given that the compute would be the right reference to use for the comparison.**
>
> Thanks for raising this point.
> The objective of Figure 1 is to show that when stripping other effects (such as IQL as the additional operator), there remains an advantage in sample efficiency from the greedification of the evaluated policy.
> To answer this question, we cannot put compute cost on the x axis.
> We agree indeed that the agents in Figure 1 result in additional compute cost (which we also mention following the figure, line 299).
> We will make sure the objective of the experiment is made more clear.
>
> **Q1. Typo:** Thanks, good catch! Corrected to "the sets are also not disjoint".
>
> **Q2. Is $I_{GMZ}$ a generalization of soft policy iteration?**
>
> Interestingly, we believe the answer is no: In soft (max. entropy) policy iteration (/SAC), we aim to get $\pi_n = softmax (q_n)$.
> That is in fact not an improvement operator outside of soft-RL (an easy proof is that $softmax (q) \neq argmax (q)$, and we know that any optimal policy is an argmax policy).
> In contrast, when we use $I_{GMZ}$ we aim to get $\pi_{n+1} = softmax (q_n + \log \pi_n)$.
> Interestingly, this *is* a policy improvement operator [4].
> In fact, this is also a limit-sufficient policy improvement operator (because $\pi_\theta$ will converge to an argmax policy that is uniform with respect to all maximizing actions, see Lemma 5 in Appendix A.6 in our paper).
>
> **Q3. non-detriment is not clear:**
>
> Thank you for pointing it out. We mean any operator that satisfies at least (in)equality 4.
> Note that this is not the same as greedification, which requires satisfying also inequality 5 (Which is the difference between the two).
> We will make sure this is made clear in the text.
>
> **Q4. Lines 282-283 are not clear**:
>
> Thank you for pointing that out.
> The first question aims to establish that the effect is there:
> that the greedification of the evaluated policy can by itself increase sample efficiency.
>
> The second question aims to establish that the effect justifies the cost:
> that it is also possible to see significant performance gains without significant increases in implementation complexity / compute.
>
> We will make sure to make it clear in the text.
>
> **Q5. It is unclear how you map the greedification operators in Alg2 to those of Alg3. When you increase the $n$ of $pg=n$, do you increase it for both the evaluated policy and acting policy?**
>
> Thanks for pointing that out.
> The increases are only to the evaluated policy (line 291).
> We will make sure that this is made more clear in the text.
> We investigate the effects of increases to the acting policy in the Appendix in Figure 6 (which is detrimental for PI while beneficial for VI).
>
> *Please let us know if there remain any unaddressed concerns, comments or questions.*
>
> [1] Huang, Audrey, et al. "Is best-of-n the best of them? coverage, scaling, and optimality in inference-time alignment." arXiv preprint arXiv:2503.21878 (2025).
>
> [2] Kostrikov, Ilya, Ashvin Nair, and Sergey Levine. "Offline reinforcement learning with implicit q-learning." arXiv preprint arXiv:2110.06169 (2021).
>
> [3] Hansen-Estruch, Philippe, et al. "Idql: Implicit q-learning as an actor-critic method with diffusion policies." arXiv preprint arXiv:2304.10573 (2023).
>
> [4] Danihelka, Ivo, et al. "Policy improvement by planning with Gumbel." International Conference on Learning Representations. 2022.

---

> > ### Comment · Reviewer_HLEH · 2025-08-02
> >
> > Thank you for your detailed and helpful rebuttal.
> >
> > > TLDR: experiments with other value-improvement operators show that signif.
> >
> > Acknowledged - I think this set of experiments gets the point across and are a good addition to the paper.
> >
> > > If we understand correctly (please correct us otherwise), the concern is that the improvement in the value is doing all the work
> >
> > Correct.
> >
> > > We first note that if the greedy operator is used the VI-GPI collapses to Value Iteration ...
> >
> > > Specifically in relation to the implicit improvement of the expectile - note that we use it with respect to freshly sampled actions from the acting policy $\pi_\theta$, not with respect to actions stored in the replay buffer.
> >
> > Ok, so perhaps my question would be (this also relates to the previous point again) about "online IQL", where there is no explicit acting policy, just one that acts greedily w.r.t. the Q function learned by IQL from replay buffer actions.
> > What are your thoughts about this / did you try comparing with such an algorithm?
> >
> > > The objective of Figure 1 is to show that when stripping other effects (such as IQL as the additional operator), there remains an advantage in sample efficiency from the greedification of the evaluated policy.
> >
> > > We agree indeed that the agents in Figure 1 result in additional compute cost (which we also mention following the figure, line 299)
> >
> > But did you try increasing the update-to-data ratio of the non-vi algorithm (td3) to the point where it gets the same compute budget as that of the vi variants? If not, how do you conclude it is more sample efficient?
> >
> > > Interestingly, we believe the answer is no: In soft (max. entropy) policy iteration
> >
> > Perhaps my terminology was incorrect, I meant policy iteration where you use a softmax in place of the max. I.e., policy mirror descent with negative-entropy regularization.

---

> ### Author Response · Authors · 2025-08-02
>
> Dear reviewer,
>
> Thank you for acknowledging our rebuttal and clarifying the remaining questions.
>
> >Ok, so perhaps my question would be (this also relates to the previous point again) about "online IQL", where there is no explicit acting policy, just one that acts greedily w.r.t. the Q function learned by IQL from replay buffer actions. What are your thoughts about this / did you try comparing with such an algorithm?
>
> In order to act greedily with respect to a Q function (learned by IQL with respect to a replay buffer, for example), one would have to search for $argmax_{a \in A} Q(s,a)$ in the continuous action space $A$ in order to act with these actions.
> In discrete actions we can find the $argmax$ by comparing the values of all actions, but in continuous  actions we need other tools.
>
> The most popular approach is that of actor critic: parameterize a function $\pi_\theta$, and search for $\pi_\theta(s) \approx argmax_{a \in A} Q(s,a)$ with gradient descent on $\theta$.
> This is exactly the policy improvement of policy gradient, and such an agent is exactly VIAC.
>
> In contrast, a more "traditional" online-IQL agent will not act greedily with respect to the $Q$ critic (learned by IQL), but attempt to extract the policy implicitly evaluated by IQL, and act with it.
> Unlike VIAC, this agent, among other possible differences, will not converge to the optimal policy unless: 1. $\tau \to 1$ and 2. The maximizing action for the optimal policy for each state appears in the replay buffer.
>
> Does this reply address the setup the reviewer has in mind?
>
> > But did you try increasing the update-to-data ratio of the non-vi algorithm (td3) to the point where it gets the same compute budget as that of the vi variants? If not, how do you conclude it is more sample efficient?
>
> Thanks for clarifying the question, we now better understand the reviewer's perspective on this Figure. Yes, in fact, we were wondering the same, and this question is investigated in Figure 7. TLDR Figure 7: when thinking of compute (gradient steps) as a resource, and asking whether to spend it on VI or on RR, Figure 7 suggests that for small increases RR results in better performance than VI.
> For larger increases the script flips, RR destabilizes faster, and VI is the better performer.
>
> We stress again that the aim of Figure 1 is simply to answer - *can sample efficiency be increased by greedifying the evaluated policy directly?*
>
> Please let us know if you find Figure 1 in its current form not suited to answering the above question.
> We will make sure the purpose of Figure 1 is made more clear in the text, and that it does not suggest that VI outperforms an increase in RR, when one considers using specifically PG for VI (which is not the method we recommend: IQL and BON show much better performance for negligable cost).
>
> > Perhaps my terminology was incorrect, I meant policy iteration where you use a softmax in place of the max. I.e., policy mirror descent with negative-entropy regularization.
>
> There is a subtle but major difference between the entropy-regularized objective $\pi_{n+1} = softmax (q_n)$, and that of $I_{GMZ}: \pi_{n+1} = softmax (\log \pi_{n} + q_n)$.
>
> The operator $ \pi_{n+1} = softmax (\log \pi_{n} + q_n) =  softmax (\log \pi_0 + \sum_{i=1}^n q_i) $ (proven in Appendix A.5).
>
> Looking at what happens when $q_n = q$, i.e. the value is stationary, perhaps sheds more light on the effect of the difference between the operators.
> In that case $ \pi_{n+1} = softmax (\log \pi_{n} + q) = softmax (\log \pi_0 + n q) $.
> If we ignore $\pi_0$ (or if it is uniform), this is very similar to the regular softmax operator: $\pi_n = softmax (\beta q)$, if the inverse temperature $\beta_n = n$ is increasing linearly over iterations. At the limit, this converges to the hard argmax (A.5), while the softmax remains a softmax regardless of $n$.
>
> More generally when the $q_n$ is not stationary, the logits of $\pi_n$ "accumulate" the information from all the previous $q_{i < n}$, concentrating over time to the action(s) with the largest sum: the maximizing actions for the sequence. If the sequence $q_n$ converges for some $q$, $\pi_n$ will also converge, and to the $argmax_a (q)$.
>
> *Please let us know if the above answers do not fully address the reviewer's questions or concerns and if there remain any additional questions or concerns.*

---

> > ### Comment · Reviewer_HLEH · 2025-08-03
> >
> > Thank you for the reply and clarifications.
> >
> > > In order to act greedily with respect to a Q function ... This is exactly the policy improvement of policy gradient, and such an agent is exactly VIAC.
> >
> > Got it.
> >
> > > In contrast, a more "traditional" online-IQL agent will not act greedily with respect to the $Q$ critic (learned by IQL), but attempt to extract the policy implicitly evaluated by IQL, and act with it. Unlike VIAC ... replay buffer.
> >
> > How do you extract the policy implicitly evaluated by IQL? If there is a standard method to do this, is this not an additional reasonable baseline? Are you arguing it is not because from a theoretical standpoint it does not converge to the optimal policy?
> >
> > > Yes, in fact, we were wondering the same, and this question is investigated in Figure 7.
> >
> >
> > I see. Figure 7 indeed answers my question.
> >
> > > We stress again that the aim of Figure 1 is simply to answer - can sample efficiency be increased by greedifying the evaluated policy directly?
> >
> > I understand. In terms of presentation, putting Figure 1 first naturally raises the question as to where the improvement is actually coming from, as greedifying the evaluated policy entails a higher utd. This makes the reader think you have perhaps instead answered the question "can sample efficiency be increased by using more compute per environment step?". Personally, I would consider a more convincing first experimental result to be one that demonstrates that extra compute is better utilized by VIAC compared to standard AC (when holding the sample count fixed), or that extra sample count is better utilized by VIAC compared to standard AC (when holding compute fixed).
> > I think Figure 7 establishes the former. I am not sure whether Fig 1 and 7 should be united or if 7 should be simply be referenced earlier in Section 4 (I now see it is referenced in the end of Section 4).
> >
> > > There is a subtle but major difference between the entropy-regularized ...
> >
> > I wasn't referring to an entropy regularized objective, I was referring to (negative-)entropy regularized policy mirror descent.
> > In any case, I think that the answer is yes, from what you write, these look the same (see e.g. Eqs 39 and 41 in [1]).
> >
> > ### Refs
> > [1] Xiao, Lin. "On the convergence rates of policy gradient methods." Journal of Machine Learning Research (2022)

---

> > > ### Author Response · Authors · 2025-08-03
> > >
> > > Dear reviewer,
> > >
> > > Thank you for the further questions and clarifications.
> > >
> > > > How do you extract the policy implicitly evaluated by IQL? If there is a standard method to do this, is this not an additional reasonable baseline? Are you arguing it is not because from a theoretical standpoint it does not converge to the optimal policy?
> > >
> > > In the original IQL paper, they use advantage weighted regression (AWR) with respect to actions in the replay buffer $D$:
> > > $ L(\theta) = E_{(s,a) \sim D}exp(\beta(Q_\phi(s,a) - V_\psi(s)))\log \pi_\theta(a|s) $.
> > > AWR can certainly be used to drive policy learning, like TD3's PG.
> > >
> > > If we understand the reviewer correctly, this thread specifically relates to the question *is the improvement to the acting policy important in any way, or is the VI doing all the work?*
> > >
> > > With this in mind, we have implemented two variations of (VI-)TD3:
> > >
> > > 1. VI-TD3, with AWR instead of TD3's deterministic PG for policy improvement, to evaluate the effect of different policy improvement operators with the same VI.
> > >
> > > 2. TD3 with the full IQL setup (V critic, expectile loss with respect to actions in the replay on the V critic, Q critic trained with targets $r(s,a) + \gamma V(s')$), and AWR in place of TD3's PG. In essence: an online IQL version which compares to TD3 on the same grounds, without changing the type of policy, architecture, etc.
> > >
> > > Preliminary results are negative: the agents struggle to compare even to the TD3 baseline, not to mention VI-TD3.
> > >
> > > We note however that an online version of IQL is not per-se an existing baseline:
> > > in both the original IQL paper [1] as well as a followup paper by the authors which explicitly analyzes IQL as an AC [2] the authors present results and agent-variations for IQL only in the offline and "online-finetuning" setups.
> > > In online-finetuning, IQL is not compared to standard online-RL methods, but only to other online-finetuning methods, where it is also not the best performer.
> > > Perhaps this indicates that a pure-online variation of IQL was not very successful.
> > >
> > > We will endeavor to include the results for agents 1. and 2. in the rebuttal and make sure to include them in the revised version of the paper whether they make it in time for the rebuttal or not.
> > >
> > > > I understand. In terms of presentation, putting Figure 1 first naturally raises the question as to where the improvement is actually coming from, as greedifying the evaluated policy entails a higher utd. This makes the reader think you have perhaps instead answered the question "can sample efficiency be increased by using more compute per environment step?". Personally, I would consider a more convincing first experimental result to be one that demonstrates that extra compute is better utilized by VIAC compared to standard AC (when holding the sample count fixed), or that extra sample count is better utilized by VIAC compared to standard AC (when holding compute fixed). I think Figure 7 establishes the former. I am not sure whether Fig 1 and 7 should be united or if 7 should be simply be referenced earlier in Section 4 (I now see it is referenced in the end of Section 4).
> > >
> > > Thanks for clarifying. We will make sure Figure 7 and the results with BON are referenced early and in relation to Figure 1.
> > > We will also make sure the interpretation of Figure 1 as similar to increasing UTD is made when the figure is discussed, and that the only conclusion that the Figure intends to convey is that *the greedification of the evaluated policy* (even when this policy is discarded after every value update, as is done in Figure 1) *can directly increase performance (measured in sample efficiency)*.
> > >
> > > If the reviewer would like additional / different changes to the presentation, please let us know.
> > >
> > > > I wasn't referring to an entropy regularized objective, I was referring to (negative-)entropy regularized policy mirror descent. In any case, I think that the answer is yes, from what you write, these look the same (see e.g. Eqs 39 and 41 in [1]).
> > >
> > > We believe the reviewer is correct and the misunderstanding is entirely ours.
> > > Equation 39 is indeed (almost) exactly what we call $I_{GMZ}$ in the paper (and exactly like the reviewer originally pointed out, $I_{GMZ}$ only generalizes further with $\sigma(q)$).
> > >
> > > We thank the reviewer for pointing this out.
> > > We will make sure to add this connection to the paper:
> > > It strengthens the theory by showing that standard operators - not limited to a specific model based algorithm, GMZ - are within the set of (limit-)sufficient greedification operators identified in our work, further demonstrating the relevance and generality of the theoretical results in Section 3.
> > >
> > > [1] Kostrikov, Ilya, Ashvin Nair, and Sergey Levine. "Offline reinforcement learning with implicit q-learning." arXiv preprint arXiv:2110.06169 (2021).
> > >
> > > [2] Hansen-Estruch, Philippe, et al. "Idql: Implicit q-learning as an actor-critic method with diffusion policies." arXiv preprint arXiv:2304.10573 (2023).

---

> > > > ### Comment · Reviewer_HLEH · 2025-08-06
> > > >
> > > > Thank you for your response and further clarifications.
> > > >
> > > > I will wait for the reviewers-AC discussion period - but am inclined to raise my rating, favoring acceptance.

---

> > > > > ### Author Response · Authors · 2025-08-06
> > > > >
> > > > > We thank the reviewer again for the fruitful discussion and the many questions and comments, which will certainly improve the paper.
> > > > >
> > > > > We report below the results for different variations of the agents discussed above.
> > > > >
> > > > > *TLDR: the improvement to the acting policy is very important and strongly influences the learning of the agent.*
> > > > >
> > > > > Neither TD3-based agent with AWR was comparable even to baseline TD3, across multiple seeds and different environments.
> > > > > For example, in hopper-stand the final performance for agent 1. was $\approx16$, and of agent 2. was $\approx6$, while TD3's was $\approx670.9$ and VI-TD3's was $\approx936.4$.
> > > > >
> > > > > Although by itself this already indicates that the improvement type to the acting policy is important, TD3 also exercises a set of design choices specifically for its constant-STD Gaussian (such as clipped sampling and different STDs for action selection and value propagation).
> > > > > It seems possible that TD3 and AWR are simply not very compatible.
> > > > >
> > > > > To verify that this finding extends beyond issues of combining TD3 with AWR we've implemented two similar variations on top of (VI-)SAC, which does learn a full Gaussian (mean and STD):
> > > > >
> > > > > 1. Identical to VI-SAC (expectile), but replaces the policy training to AWR with respect to the actions in the replay buffer.
> > > > >
> > > > > 2. A full "soft-online-IQL" agent: replaces the policy training to AWR with respect to the actions in the replay buffer, like agent 1., and second, replaces the expectile loss to the original manner (with respect to actions from the replay buffer). This, in contrast to ours, which is with respect to fresh actions from the policy (and is used by Agent 1.).
> > > > >
> > > > > We include here results across three environments which we've chosen randomly from the environments where there were significant performance difference between SAC and VI-SAC (Figure 3).
> > > > >
> > > > > We report final performance after $3m$ environment interactions + SEM, averaged across 20 seeds for SAC and VI-SAC, and 7 seeds for Agents 1. and 2. (the number of seeds we were able to run within the confines of the rebuttal).
> > > > >
> > > > > Neither agent 1. nor agent 2. are competitive with the SAC baseline, and certainly not with VI-SAC.
> > > > > We conclude that the improvement to the acting policy is a major influential component in VI-AC algorithms.
> > > > >
> > > > > | Task              | SAC | VI-SAC (expectile) | Agent 1. | Agent 2. |
> > > > > |-|-|-|-|-|
> > > > > | hopper-stand      | 591.7 ± 82.3       | **939.4 ± 3.8**| 25.7 ± 14.3        | 144.7 ± 92.4       |
> > > > > | hopper-hop        | 97.8 ± 17.0        | **296.0 ± 15.8**| 0.4 ± 0.4          | 0.0 ± 0.0          |
> > > > > | humanoid-run      | 62.6 ± 15.8        |**126.8 ± 8.1**| 0.7 ± 0.2          | 0.3 ± 0.2          |
> > > > >
> > > > > We will include in the paper results for these agents in the form of learning curves, to more comprehensively demonstrate the difference in learning behaviors.

---

### Official Review · Reviewer_TiJB · 2025-07-02

**Clarity:** 3
**Significance:** 2
**Originality:** 3
**Rating:** 4
**Confidence:** 3

**Summary:**

This paper introduces a new framework called Value-Improved Actor-Critic (VIAC), which separates the policy that interacts with the environment from the one being evaluated by the critic. The main idea is to let the critic evaluate a more "greedy" policy to get better value estimates, while the actor still updates slowly and stably using gradients. The authors back this up with a solid theoretical analysis—showing under what conditions this separation still leads to convergence—and test the idea both explicitly (with multiple critic-side updates) and implicitly (using expectile loss). They apply it to TD3 and SAC and find that the implicit version, in particular, gives nice performance gains with almost no extra compute.

**Questions:**

1. Could the authors respond to the concerns raised in the weaknesses section, particularly the first point regarding whether the implicit method genuinely performs value improvement?

2. Could the authors elaborate on how the mismatch between the actor and critic policies affects learning? For example, if the two policies diverge significantly, does this impact training stability or sample efficiency?

3. The theoretical results are grounded in finite MDPs, but modern deep RL involves continuous state and action spaces, along with approximation error. How well do the guarantees hold up under function approximation and in the presence of non-stationary Q estimates?

4. The balance between greedy updates and stable improvement somewhat reminds me of TRPO, which deliberately avoids overly greedy policy updates. Do the authors see any conceptual connection here? It would be interesting to hear their thoughts.

**Ethical Concerns:**

["NO or VERY MINOR ethics concerns only"]

**Final Justification:**

The rebuttal resolved my main concerns: IQL’s fit within VIAC is now clear, bias analysis shows gains beyond estimation changes, and BON results confirm practical value. Only minor uncertainty remains on theory under deep RL. Given these, I am raising my score.

**Limitations:**

Yes.

**Paper Formatting Concerns:**

no.

**Quality:**

3

**Strengths And Weaknesses:**

**Strengths:**

1. The paper presents a novel and elegant way to address a long-standing tradeoff in actor-critic methods: the tension between greediness and stability.
2. The theoretical analysis is thorough and introduces meaningful distinctions between types of policy improvement operators, offering a clear view into when convergence is (or isn’t) guaranteed. I also like the intuition for the theories.
3. The empirical results are compelling. The authors show that integrating value improvement into TD3 and SAC leads to consistent performance gains across a range of continuous control tasks.
4. The implicit implementation using expectile loss is particularly practical—it’s simple to implement, introduces negligible overhead, and still captures the benefits of value improvement.

**Weaknesses:**

1. I’m not entirely convinced that the implicit version using expectile loss fits neatly within the VIAC framework. There is an important distinction between genuinely greedy value estimation and simple overestimation—larger value estimates may stem from estimation bias rather than deliberate greedification. It remains unclear whether the observed improvements are due to a shift toward a greedier policy or simply the result of over- or underestimation. This is my primary concern, and I would appreciate a more detailed explanation from the authors on this point.
2. While the explicit variant is mainly included to motivate the framework conceptually, its high computational cost makes it impractical for most real-world use. That wouldn’t be a problem on its own—after all, the implicit version is the one intended for practical use. However, given my concern above about whether the implicit method truly reflects value improvement in the VIAC sense, the lack of a practical, clearly greedy implementation slightly weakens the overall strength of the framework. If expectile loss isn’t truly implementing a greedy operator, then the framework may not be as broadly applicable in practice as it first appears.
3. While some of the evaluated environments (e.g., cartpole-swingup, hopper-stand) exhibit partially sparse rewards, the benchmark suite is still relatively standard. It would be interesting to test the method in more extreme sparse-reward settings, such as long-horizon navigation or delayed-reward tasks, to further validate its robustness.



Overall, I found the paper clear and well-motivated. If I misunderstood anything, I'm happy to be corrected, and I’m open to revisiting my evaluation based on the authors’ responses.

---

> ### Author Rebuttal · Authors · 2025-07-28
>
> Dear reviewer,
>
> Thank you for your time, thoughtful review and detailed comments and questions.
>
> **1a. Does IQL fit within VIAC?**
>
> *TLDR: We conclude that yes, the expectile operator fits within the VIAC framework.*
>
> [1] prove that the expectile loss explicitly evaluates greedier policies with respect to some sampling policy $\pi_b$.
> In their setting of offline RL, $\pi_b$ is captured as the distribution over the actions per state in the replay buffer.
> However, since we are not in the offline RL setting, we use the expectile loss with respect to fresh actions from $\pi_\theta$ directly (see Appendix D.3), which results in an operator that explicitly greedifies with respect to $\pi_\theta$.
> Further, [2] show that the policy $I_{IQL}(\pi, Q)$ can be extracted explicitly.
> We conclude that when $ \tau > 0.5: \sum_{a \in A} I_{IQL}(\pi, Q)(a|s) Q(s,a) > \sum_{a \in A} \pi(a|s) Q(s,a) $, i.e., the expectile loss results in explicit greedification, and by using it with respect to fresh actions from $\pi_\theta$, it explicitly greedifies with respect to $\pi_\theta$.
>
> The expectile operator however will not converge to the greedy policy unless $\tau \to 1$, and for that reason, it is not a *sufficient* improvement operator unless $\tau \to 1$.
> This raises again the question whether IQL fits within the VIAC framework or if it conflict with the theory. Corollary 2 addresses this by proving that even when insufficient ("non-detriment") operators are used for value improvement, the algorithm still converges.
>
> **1b. It remains unclear whether the observed improvements are due to a shift toward a greedier policy or simply the result of over- or underestimation.**
>
> *TLDR: We conclude that the results presented in Figures 1 & 4 & additional results included here, suggest that performance gains of VIAC go beyond simply the result of over or underestimation.*
>
> In Figure 1 (second row, right) we see performance improvement without change in estimation bias.
> This shows that performance gains with VIAC can go beyond simply the result of over or underestimation.
>
> Overall, we would expect to see 3 behaviors, differing across environments:
> (i) Where some overestimation is beneficial, VIAC may benefit from increased greedification up to the point of over-estimation increase. As long as over estimation is beneficial, it can benefit from the over estimation induced by increased greedification.
> (ii) Where overestimation is mostly detrimental, VIAC may benefit from increased greedification up to the point of over-estimation increase. Additional greedification will tradeoff against any negative effects of overestimation.
> (iii) As long as the effects of overestimation are minor, VIAC may benefit from increased greedification irrespective of overestimation.
>
> In Figure 4, we see that humanoid-run belongs essentially to (iii): the entire performance improvement is experienced when greedifying $\tau \to 0.6$, up to which point the estimation bias is not strongly influenced.
> From $\tau > 0.6$, the estimation bias monotonically increases while performance remains roughly the same, until a point where the overestimation is too high and performance decays.
>
> We include a table which extends and summarizes Figure 4 to all environments.
> We report:
>
> (i) The maximum final performance increase (final performance - baseline final performance) observed over $\tau$s, without (<1%) increase in overstimation. The $\tau$ associated with this final performance.
>
> (ii) The maximum final performance increase observed over $\tau$s. The associated $\tau$. The % overestimation increase out of the maximum overestimation observed in this experiment. The % overestimation / performance for this $\tau$.
>
> Averaged across 10 seeds per $\tau$, and along with SEM. Stat. signif. differences from baseline are marked in bold.
>
> In 5 environments (hopper-hop, humanoid-stand/walk/run, acrobot-swingup) there is significant performance gain without an increase in over estimation.
> In 6 environments (hopper-stand, quadruped-walk/run, fish-swim, walker-walk/run) any performance gain is correlated with some increase in over estimation.
> In 1 environment (cheetah-run) there is no statistically significant difference between the baseline and VI-TD3.
>
> We conclude that the benefits of VIAC go beyond increasing over estimation bias.
> On the other hand, VIAC can be used to increase estimation bias, and when it is beneficial, VIAC can benefit from it (and when it is not beneficial, it's one of the reasons it is important that the amount of greedification be tuned).
>
> |Environment|Gain w.o. overestimation increase|$\tau$|Max. perf. gain|$\tau$|Overestimation % of max. overestimation|Overestimation as % of perf.|
> |-|-|-|-|-|-|-|
> |hopper-stand|-|-|353.80 ± 60.77|0.75|5%|2%|
> |hopper-hop|88.01 ± 24.31|0.59|217.25 ± 26.38|0.8|12%|2%|
> |humanoid-stand|206.79 ± 64.93|0.55|503.93 ± 14.72|0.75|31%|5%|
> |humanoid-walk|320.64 ± 43.42|0.6|415.28 ± 29.29|0.7|14%|3%|
> |humanoid-run|107.50 ± 5.38|0.6|125.25 ± 3.78|0.75|22%|5%|
> |quadruped-run|-|-|166.18 ± 47.69|0.6|3%|1%|
> |quadruped-walk|-|-|276.55 ± 59.85|0.59|1%|1%|
> |fish-swim|-|-|308.07 ± 19.02|0.9|87%|4%|
> |walker-walk|-|-|27.04 ± 9.63|0.7|28%|1%|
> |walker-run|17.65 ± 29.07|0.53|98.23 ± 25.79|0.75|27%|1%|
> |cheetah-run|10.37 ± 11.18|0.53|10.37 ± 11.18|0.53|4%|1%|
> |acrobot-swingup|22.74 ± 3.94|0.6|257.24 ± 6.13|0.8|11%|6%|
>
> **2. Practical value improvement with other, low cost, explicit greedification operators:**
>
> *TLDR: In additional results we include here other - low cost - explicit greedification operators demonstrate significant  and comparable increases to performance.*
>
> We include an example of an explicit improvement operator that is inexpensive and shows strong performance benefits in these domains: Best-of-n (BON). BON samples N actions, takes the value of the maximizing action as the bootstrap for the value target.
> This is a popular operator in RL for language models [3].
> We include a table that summarizes a figure like Figure 3 for baseline TD3, and the IQL and BON variations of VI-TD3.
> $N=64$ for BON and for IQL we use the same $\tau=0.75$.
>
> BON shows similar performance gains to IQL across domains.
>
> Below we report final-performance results normalized per environment between minimum and maximum final return over all seeds (20 for TD3 and VI-TD3 (IQL), and 12 for VI-TD3 (BON)), averaged per agent over environments, with +- one SEM.
> The IQL and BON variations are within each others 2 SEM confidence bound.
>
> |Agent|Normalized Final Perf.|
> |-|-|
> |Baseline TD3|0.41 ± 0.05|
> |VI-TD3 (IQL)|0.75 ± 0.04|
> |VI-TD3 (BON)|0.68 ± 0.04|
>
> For results per env., due to limited space, see the reply to reviewer HLEH.
>
> **3. Sparse rewards:**
>
> Due to the possibility of explicitly tuning over-estimation bias through the tuning of the greedification with VIAC, we would expect VIAC to be able to induce pseudo-optimism-in-the-face-of-uncertainty exploration which should perform even better than baselines that do not have it, in sparse reward domains.
>
> **Q2. Could the authors elaborate on how the mismatch between the actor and critic policies affects learning?**
>
> In our experiments, we have only tried operators that sample from the acting policy $\pi_\theta$ and thus do not majorly diverge from it (like BON or IQL). We would conjecture that the important factor is whether the critic predictions are reliable for the evaluted policy. if OOD actions are evaluated, we would expect off-policy-evaluation concerns.
>
> **Q3. The theoretical results are grounded in finite MDPs, but modern deep RL involves continuous state and action spaces, along with approximation error. How well do the guarantees hold up under function approximation and in the presence of non-stationary Q estimates?**
>
> *Cont. actions*: While this is not usually analyzed in the setup of DP (to our knowledge), our theory is not fundementally limited to discrete actions. The induction does not depend on the action space, only on whether the improvement operators can greedify the policy all the way to an argmax.
>
> *Approximation error*: When taking specific operators (say policy gradient for PI and the argmax for VI), under specific function approximation assumptions (presence of local optima, bounds on the possible error, etc), we don't see a reason to assume VIAC will not converge: VIAC simply maintains a Q estimate that is larger at any step than that of AC, but is still anchored in real policies.
>
> Note that our analysis is already in the presence of non-stationary and approximate Q estimates. Similarly, we conjecture that the results can generally extend to other standard methods to approximate Q, such as Monte-Carlo / TD-$\lambda$ targets.
>
> **Q4. The balance between greedy updates and stable improvement somewhat reminds me of TRPO, which deliberately avoids overly greedy policy updates. Do the authors see any conceptual connection here?**
>
> Yes absolutely.
> We touch on a similar connection in the introduction, in lines 43-49.
> If we think of the improved policy of TRPO $\pi'$ as the "optimally greedy" policy, the evaluated policy is still limited to $\pi_\theta$, which is only moved towards $\pi'$ with gradient steps.
> VIAC in contrast would suggest directly evaluating $\pi'$, which seems very natural for TRPO. Does that make sense?
>
> *All results presented in the rebuttal will be included in the appendix.*
>
> *Please let us know if there remain any unaddressed concerns, comments or questions.*
>
> [1] Kostrikov, Ilya, Ashvin Nair, and Sergey Levine. "Offline reinforcement learning with implicit q-learning." arXiv preprint arXiv:2110.06169 (2021).
>
> [2] Hansen-Estruch, Philippe, et al. "Idql: Implicit q-learning as an actor-critic method with diffusion policies." arXiv preprint arXiv:2304.10573 (2023).
>
> [3] Huang, Audrey, et al. "Is best-of-n the best of them? coverage, scaling, and optimality in inference-time alignment." arXiv preprint arXiv:2503.21878 (2025).

---

> > ### Author Response · Authors · 2025-08-06
> >
> > Dear reviewer,
> >
> > Thank you again for the detailed review and many comments and questions.
> > Please let us know if you have any concerns or questions remaining after our rebuttal.

---

> > ### Comment · Reviewer_TiJB · 2025-08-09
> > **Thank you for your response**
> >
> > Thank you for the detailed reply. The rebuttal addresses most of my concerns. The clarification on IQL within VIAC, the extended bias analysis, and the BON results strengthen both the theoretical and practical case. Novelty concerns are partly alleviated, and empirical evidence supports benefits beyond bias changes. Remaining reservations on theory under deep RL remain minor. I am updating my score.

---

### Decision · Program_Chairs · 2025-09-17

**Decision:**

Accept (poster)

**Comment:**

This paper introduces Value-Improved Actor-Critic (VIAC), a general framework that decouples the acting policy from the evaluated policy in actor-critic algorithms. The central idea is to apply a greedier operator when evaluating the policy (value improvement), while maintaining the slower, gradient-based updates for the acting policy to ensure stability. This separation directly addresses the long-standing tradeoff between greedification and stability in actor-critic methods. The framework is grounded in generalized policy iteration, with convergence guarantees in finite-horizon settings. Empirical results show that incorporating VIAC into TD3 and SAC leads to consistent improvements across multiple continuous control benchmarks, with negligible additional computational cost.

The reviewers raised thoughtful questions about whether the implicit version (based on expectile loss) truly implements value improvement, or whether its gains stem from estimation bias. The authors’ rebuttal addressed this by demonstrating that improvements are not solely due to over/underestimation (e.g., performance gains occur even without bias change) and that value and policy improvement complement each other rather than rendering one another redundant. While some concerns remain about the simple discrete state-action space setting, limited scope of experiments (e.g., sparse reward, high-dimensional inputs) and the practicality of the explicit variant, all reviewers agree that the technical contribution is novel, well-motivated, and supported by both theoretical analysis and empirical evidence.